# Reconstitution of human PDAC using primary cells reveals oncogenic transcriptomic features at tumor onset

Yi Xu[1,9], Michael H. Nipper[1,9], Angel A. Dominguez [1], Zhenqing Ye [2,3], Naoki Akanuma [1], Kevin Lopez[1], Janice J. Deng [1], Destiny Arenas[1], Ava Sanchez [1], Francis E. Sharkey [4], Colin M. Court [5], Aatur D. Singhi[6], Huamin Wang [7], Martin E. Fernandez-Zapico [8], Lu-Zhe Sun [1], Siyuan Zheng [2,3], Yidong Chen[2,3], Jun Liu[1] ✉ & Pei Wang [1] ✉

Animal studies have demonstrated the ability of pancreatic acinar cells to transform into pancreatic ductal adenocarcinoma (PDAC). However, the tumorigenic potential of human pancreatic acinar cells remains under debate. To address this gap in knowledge, we expand sorted human acinar cells as 3D organoids and genetically modify them through introduction of common PDAC mutations. The acinar organoids undergo dramatic transcriptional alterations but maintain a recognizable DNA methylation signature. The transcriptomes of acinar organoids are similar to those of disease-specific cell populations. Oncogenic *KRAS* alone do not transform acinar organoids. However, acinar organoids can form PDAC in vivo after acquiring the four most common driver mutations of this disease. Similarly, sorted ductal cells carrying these genetic mutations can also form PDAC, thus experimentally proving that PDACs can originate from both human acinar and ductal cells. RNA-seq analysis reveal the transcriptional shift from normal acinar cells towards PDACs with enhanced proliferation, metabolic rewiring, down-regulation of MHC molecules, and alterations in the coagulation and complement cascade. By comparing PDAC-like cells with normal pancreas and PDAC samples, we identify a group of genes with elevated expression during early transformation which represent potential early diagnostic biomarkers.

Pancreatic ducal adenocarcinoma (PDAC) is the most common histological subtype of pancreatic cancer. This malignancy is one of the deadliest human cancers and is predicted to be the second leading cause of cancer-associated death by 2040[1]. Lack of effective treatments for late stage PDAC is one of the reasons for its dismal survival rate, but major challenges also include late diagnosis, resistance to chemotherapy, and metastasis[2]. Prevention and early diagnosis represent the most promising strategies for improving outcomes

[1]Department of Cell Systems & Anatomy, University of Texas Health Science Center at San Antonio, San Antonio, TX 78229, USA. [2]Greehey Children's Cancer Research Institute, University of Texas Health Science Center at San Antonio, San Antonio, TX 78229, USA. [3]Department of Population Health Sciences, University of Texas Health Science Center at San Antonio, San Antonio, TX 78229, USA. [4]Department of Pathology and Laboratory Medicine, University of Texas Health Science Center at San Antonio, San Antonio, TX 78229, USA. [5]Division of Surgical Oncology and Endocrine Surgery, University of Texas Health Science Center at San Antonio, San Antonio, TX 78229, USA. [6]Department of Pathology, University of Pittsburgh Medical Center, Pittsburgh, PA 15213, USA. [7]Department of Pathology, University of Texas MD Anderson Cancer Center, Houston, TX 77030, USA. [8]Schulze Center for Novel Therapeutics, Division of Oncology Research, Mayo Clinic, Rochester, MN 55905, USA. [9]These authors contributed equally: Yi Xu, Michael H. Nipper. ✉e-mail: Liuj8@uthscsa.edu; wangp3@uthscsa.edu

of this disease. A better understanding of the mechanisms of early PDAC tumorigenesis is the first step towards this goal. However, our current knowledge about early PDAC progression in human cells is still very limited, with some fundamental questions remaining unclear, including what the cellular origin of PDAC is and how the normal exocrine cells transform to malignant cells during PDAC initiation in response to environmental cues and oncogenic events.

More than 90% of the pancreatic mass is exocrine tissue comprised of acini that secrete large amounts of digestive enzymes and ducts that deliver the acinar secretions to the duodenum[3]. The duct-like histological morphology of PDACs led to the belief that human PDAC was originated from ductal cells[4]. Although multiple mouse PDAC models have been developed in the past two decades[5], modeling human PDAC tumorigenesis remains challenging, as current human pancreatic organoids are mainly derived from human PDAC patients with advanced disease[6,7]. In addition, animal studies have demonstrated that both acinar and ductal cells can serve as the cellular origin for mouse PDAC and that cell-of-origin influences the molecular subtypes and aggressiveness of PDACs[8–13]. However, the cell-of-origin in human PDAC has yet to be proven.

In the present study, we established long-term culture systems for primary human pancreatic acinar and ductal cells followed by genetic engineering to reconstruct early human PDAC development. Our results provide direct evidence for both acinar and ductal origin of human PDAC formation, and reveal the effects of classical PDAC driver mutations on transcriptional reprogramming in human cellular context. In addition, we identified a group of genes upregulated during the transition stage from normal exocrine cells to early PDAC. These genes were also highly expressed in TCGA PDAC data compared with normal pancreas data, suggesting their possible participation in human pancreatic tumorigenesis as well as their potential as early diagnostic biomarkers.

## Results

### Long-term 3D acinar cells culture mimics in vivo injuries-induced metaplasia

Disease-free islet-depleted pancreatic acinar fractions were obtained from human organ donors (Supplementary Data 1) and stained for known acinar marker AMY2 and ductal marker KRT19, which revealed the presence of both acinar and ductal lineages (Fig. 1a). Flow cytometry was then used to sort the two lineages. Sorted acinar cells (UEA-1$^{high}$CLA$^-$)[14] showed distinct secretory zymogen granules and rough endoplasmic reticulum structures by electron microscopy (EM), confirming acinar identity (Fig. 1a). Sorted ductal cells (UEA-1$^{low}$CLA$^+$) showed ductal-associated cilia structures.

RNA-seq analysis of sorted cells from 5 independent donors revealed differential expression of 2087 acinar-high genes and 1906 ductal-high genes, including 273 differentially expressed transcriptional factors (TFs) (Fig. S1a, b, fold-change > 2; adj.$p < 0.05$). As expected, the most highly expressed genes in the acinar population included many digestion enzymes (Fig. S1c). The top enriched gene ontology (GO) terms in acinar-high genes included cytoplasmic translation, ribosome assembly, digestion, and protein secretion, corroborating normal acinar function (Fig. 1b). In comparison, ductal cells expressed high levels of tubule morphogenesis-associated genes and several signaling pathways including WNT pathway and transforming growth factor beta pathway.

To establish an acinar and ductal culture model, we cultured sorted cells separately in a WNT-dependent 3D organoid system[7]. Interestingly, the ductal-derived organoids initially proliferated rapidly, but ceased to proliferate within 4 passages (Fig. 1c). On the other hand, the acinar-derived organoids proliferated at a much slower pace during early passages, then started to expand rapidly and could be continually passaged for more than 6 months and cryopreserved. The cultured acinar organoids showed a typical ring-like morphology

and expression of MUC5AC, a marker for KRAS induced acinar metaplasia (Fig. S1d).

The transcriptomic profiles of acinar-derived organoids (in vitro culture for at least 6 weeks) established from 6 independent donors appeared to be distinct from both fresh acinar and fresh ductal cells (Fig. S1e). A comprehensive three-group comparison of the fresh acinar, fresh ductal, and cultured acinar organoid expression data identified >10,000 differentially expressed genes (DEGs) across the three groups (Fig. 1d, Supplementary Data 2, adj.$p < 0.05$). Analysis of the 2087 acinar signature genes revealed that the organoids lost most acinar lineage characteristics, which were mainly involved in normal acinar functions (Fig. S1f). On the other hand, the cultured organoids acquired only a portion of ductal genes including some ductal-specific transcription factors (Fig. S1g), highlighting the fact that they did not trans-differentiate into ductal lineage.

Recent mouse pancreas injury studies revealed that acinar cells could transform to multiple metaplastic populations during acinar-to-ductal metaplasia (ADM)[15–19]. Tosti et al. showed the emergence of tuft cell and Mucin5B/ductal cell populations in human chronic pancreatitis samples[20]. Many gene markers of these metaplastic populations were up-regulated in our acinar-derived organoids (such as AQP5, TFF2, MUC6, PGC, CLDN18, MUC5AC, TFF1, MUC5B, etc, Fig. 1e), suggesting that our 3D culture system mimics the in vivo pancreatic acinar injuries. To perform a gene expression comparison of our cultured acinar cells with referenced metaplastic cell populations, a pseudo-bulk RNA-seq data was created for each of these metaplastic populations by aggregating read counts from single cell RNA seq (scRNA-seq) data. The principal component analysis (PCA) showed that our cultured acinar organoids shared great similarities with chief-like cells, pit-like cells as well as Mucin5B/ductal cells identified in mouse and human pancreas disease models (Fig. 1f). Notably, the cellular origin of human pancreatitis associated Mucin5B/ductal cells was unclear, as lineage tracing is not possible in human clinical studies. However, the transcriptional similarity with our cultured organoids implies an acinar origin for this disease-associated cell type in human patients.

To investigate the potential heterogeneity of our cultured acinar cells, we performed a bulk RNA deconvolution analysis using a Multi-Subject Single Cell deconvolution (MuSiC) method[21] to infer the cell type composition in our bulk RNA seq samples. By using a mice scRNA-seq dataset[19] containing normal and metaplastic acinar cells as a reference, the deconvolution analysis inferred ~90% of our fresh acinar cells as normal acinar cells, demonstrating the robustness of the analysis (Fig. S1h). Importantly, the analysis inferred >90% of our cultured acinar cells as the metaplastic chief-like acinar cells, instead of any other cell types present in the reference dataset. The analysis thus showed a great possibility that our cultured acinar cells may resemble a specific metaplastic cell type with limited heterogeneity.

### Cultured pancreatic exocrine cells maintained recognizable lineage-associated DNA methylation features

The substantial transcriptional and morphological changes made the lineage origin of cultured organoids ambiguous. We hypothesized that DNA methylation may remain relatively stable and can be used to verify lineage identity. To this end, reduced representation bisulfite sequencing was performed to assess the DNA methylation of fresh sorted acinar and ductal cells. The correlation of DNA methylation of all analyzed CpG sites was much higher among samples of a given lineage (Fig. 2a, S2a). Differentially hypermethylated sites were more likely to be found in introns and intergenic regions (Fig. S2b, c). Among the top expressed genes in the pancreas according to the Genotype-Tissue Expression (GTEx) project, promoters of genes associated with translation machinery were hypomethylated in the fresh cells of both lineages, while those associated with pancreatic digestive enzymes were hypomethylated in fresh acinar cells and hypermethylated in fresh ductal cells (Fig. 2b). A notable exception, the insulin (INS) gene

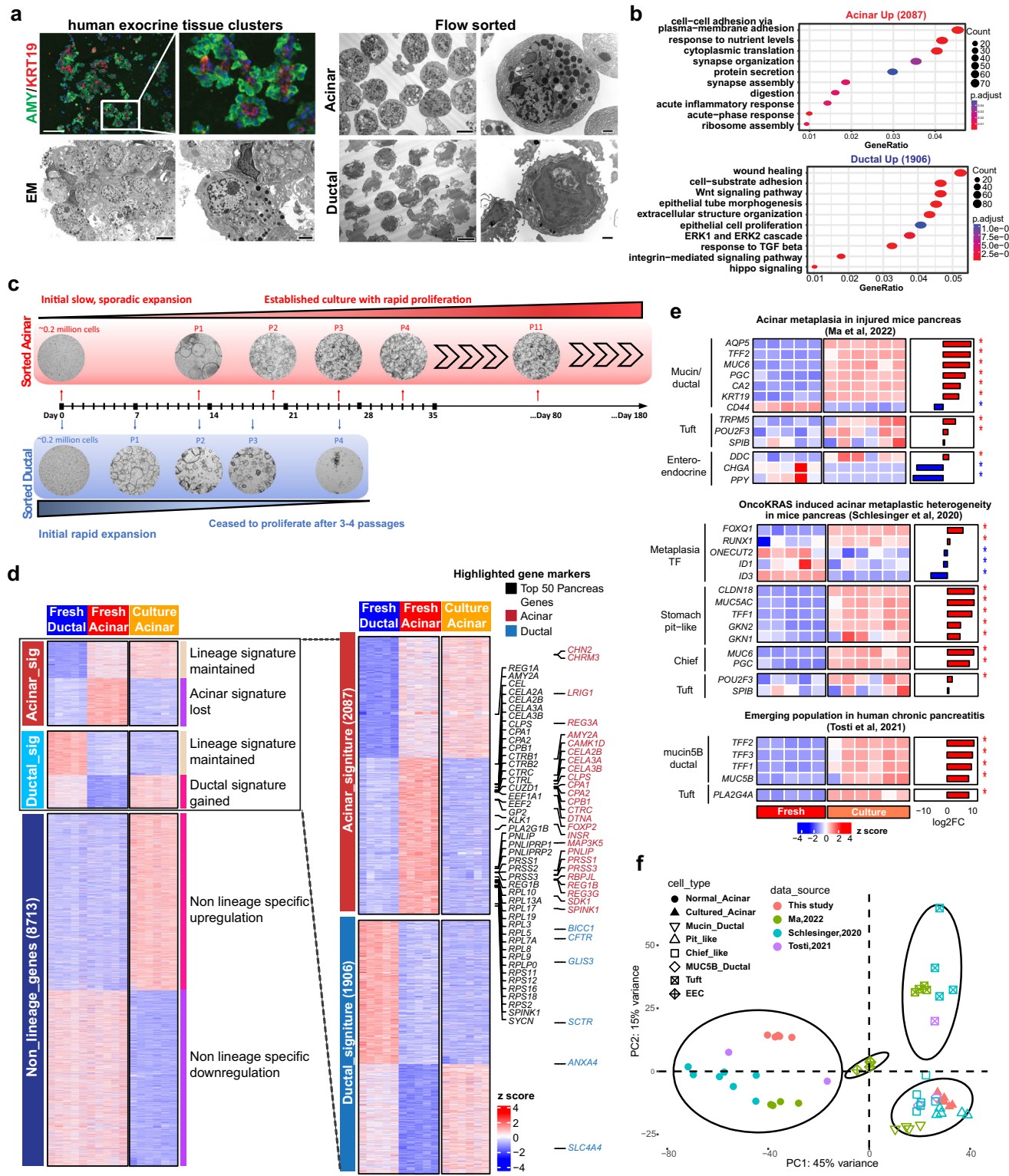

was hypermethylated in both acinar and ductal cells. Consistent with RNA expression profile, most of the highly DEGs in fresh acinar cells were hypermethylated in fresh ductal cells and vice versa (Fig. S2d).

Although long-term 3D ductal organoid culture failed, we were able to establish four independent 2D ductal cultures from two donor tissues by introducing oncogenic *KRAS* and mutations in *p16, TP53*, and *SMAD4* immediately after cell sorting. All the experiments throughout this work involving cultured or genetically modified ductal samples were all performed using 2D ductal culture. To validate the lineage authenticity of the established cultures, DNA methylation profiling was

performed for five acinar 3D and four ductal 2D independent cultures. The hierarchy clustering analysis revealed that cultured acinar and cultured ductal cells were closely related with their fresh counterparts, respectively (Fig. 2a). In addition, although the gene expressions of pancreatic digestive enzymes were lost in cultured acinar organoids, the promoter methylation patterns for many of them were maintained (Figs. 2b, S2e).

Of the 5410 fresh acinar hypomethylated sites with a beta value difference of at least 0.8, 2691 (49.7%) were maintained, 1916 (35.4%) gained methylation in cultured acinar cells, and 803 (14.8%) lost

**Fig. 1 | Primary human 3D acinar culture associated with metaplastic features.**
**a** Left: Immunofluorescence staining of Amylase and KRT19 ($n = 6$ fields of views, scale bar = 100 μm), and electron microscopy images of islet-depleted human pancreatic exocrine tissue ($n = 8$ fields of views, scale bar = 2 μm). Right: Electron microscopy images of the flow-sorted acinar and ductal cells ($n = 4$ fields of views for each lineage, scale bar = 6 μm, for enlarged images scale bar = 1 μm). **b** Gene ontology analysis of the differentially expressed genes in sorted acinar ($n = 5$) and ductal cells ($n = 5$). The analysis was performed using the clusterProfiler 4.4.4 package in R software with default settings. A significant enrichment was considered with multiple-test adjusted $p$ value < 0.05. **c** Bright field images and schematic illustration of established long-term 3D acinar culture. **d** Expression heatmap of all differentially expressed genes across 3 groups of samples including fresh acinar cells ($n = 5$), fresh ductal cells ($n = 5$), and cultured acinar organoids ($n = 6$). For three-group comparison, the DEGs were identified by likelihood ratio test using DESeq2 1.36.0 package in R software, and defined as Benjamin-Hochberg adjusted $p$ value < 0.05. For two-group comparison (fresh acinar vs fresh ductal), DEGs were identified by negative binomial Wald test using DESeq2 with fold change >2 and adjusted $p$ value < 0.05. The genes were manually grouped into 3 clusters according to their expression patterns in sorted fresh cells. Within each of the 3 gene clusters, K means clustering further identified 2 subgroups of genes with distinct expression patterns. Selected known pancreatic signature genes were annotated with colored text. **e** Expression heatmap and log2 fold changes of listed acinar metaplasia genes in cultured vs fresh acinar cells. The listed genes were curated from indicated references. To compare gene expression between cultured vs fresh acinar cells, negative binomial Wald test of the 2 groups of samples was performed using DESeq2 in R software. Asterisk indicates adj.$p$ < 0.05. **f** Principal component analysis of the gene expression in fresh and cultured acinar samples generated in present work as well as normal and metaplastic acinar populations from referenced work shown in (**e**). To compare our bulk RNA seq samples with the referenced scRNA-seq samples, a pseudo-bulk RNA seq profile was generated for each cell population in referenced scRNA-seq dataset by aggregating the gene counts of all cells in a given cell population.

methylation in cultured ductal cells (Fig. 2c). Of the 10735 fresh ductal hypomethylated sites, 5384 (50.1%) were maintained, 2894 (26.7%) gained methylation in cultured ductal cells, and 2457 (22.9%) lost methylation in cultured acinar cells (Fig. 2c). Gene Ontology analysis of the genes associated with these sites revealed that the maintained methylation patterns largely correlated with the normal functions of the cell of origin while altered DNA methylation patterns were associated with stress response. The genomic distribution of the sites within each methylation pattern was similar to the distribution of all hypomethylated sites in a given lineage, implying that changes in DNA methylation were not a result of generalized failure of DNA methylation maintenance enzymes (Fig. 2d, e). These data confirmed the lineage identity of our established cultures, which supported the conclusion that primary human acinar cells can be cultured in vitro as a model for studying pancreatic disease.

## Human primary acinar and ductal cells can both form PDAC

Acquisition of oncogenic *KRAS* mutation is the earliest and the most prevalent genetic alteration in human PDAC[22]. To recapitulate the molecular events underlying PDAC initiation, we first overexpressed oncogenic *KRAS^G12V^* in wild-type acinar organoids (referred to KRAS organoids) by lentiviral infection with a vector containing *KRAS^G12V^*-mCherry. Surprisingly, the gain of oncogenic *KRAS* only led to the identification of two DEGs (fold-change >2; adj.$p$ < 0.05) including *KRAS* and *ST3GAL1*, a known mediator of metabolic reprograming driven by oncogenic *KRAS* in mice PDAC[23] (Fig. 3a). Interestingly, we found that the unmodified wild-type acinar organoids exhibited elevated KRAS-induced PDAC transcription signatures identified in mouse models, such as glycolysis, steroid biosynthesis, and O-glycan biosynthesis pathways[23] (Fig. 3b). In addition, the wild-type acinar organoids expressed MUC5AC at a high level independently of oncogenic *KRAS* (Fig. S1d), while induction of MUC5AC expression in mouse ADM required oncogenic *KRAS*[15]. It is possible that supplements such as epidermal growth factor (EGF) in our culture medium may substitute for oncogenic *KRAS* to induce pre-cancer signatures in organoid culture. Nevertheless, these KRAS organoids failed to form tumors when transplanted into in NSG (NOD.Cg-*Prkdc^scid^Il2rg^tm1Wjl^*/SzJ) mice subcutaneously.

We next knocked out *p16/CDLN2A* (P), *TP53* (T) and *SMAD4* (S) genes in acinar KRAS organoids using CRISPR/Cas9 system, and compared with ductal-KPTS 2D cultures (Figs. 3c, S3a, b). RNA-seq analysis of three independent acinar-KPTS 3D cultures and four ductal-KPTS 2D cultures identified of >4000 DEGs (adj.$p$ value < 0.05, fold change >2) between KPTS cells derived from two lineages (Fig. S3c). Notably, by comparing with a lineage-specific mouse PDAC model[12], we identified a group of genes significantly highly expressed in both human acinar-KPTS cells and mouse acinar-derived tumors, including *MUC5AC, TFF1, TFF2, CLDN18*, and *REG1A*

(Fig. 3d, Supplementary Data 3). Similarly, a group of genes were identified that are significantly highly expressed in both human ductal-KPTS cells and mouse ductal-derived tumors, including *EREG, WNT7B, EDL3, GATA3* and *KCNAB1* (Fig. 3d). In addition, by comparing the DEGs between two lineages at fresh state and at KPTS state, we identified a list of genes whose lineage-associated expression pattern was maintained even after genetic modifications (Fig. S3d, Supplementary Data 4). To assess the heterogeneity among our samples from different donors and culture states, PCA analysis was performed including the fresh, cultured, genetically modified acinar and ductal samples collected in this work. Result showed that the samples clustered according to their respective culture states, as opposed to different donor origins (Fig. 3e).

The lack of direct evidence for acinar transformation in human disease prompted us to transplant the acinar and ductal KPTS cultures subcutaneously into immunodeficient NSG mouse to assess their tumorigenicity. Results showed that tumors can be reproducibly generated from both lineages harboring oncogenic KPTS mutations (acinar, $n = 15$, ductal, $n = 30$, Fig. 3f). We noted that ductal-derived tumors were generally larger than acinar tumors. Histological analysis revealed that both the acinar and ductal-derived tumor tissues had duct-like structures with intraluminal papillae and prominent hyalinized or desmoplastic stroma, resembling early human PDAC morphology (Fig. 3f). In addition, typical pancreatic intraepithelial neoplasia (PanIN) lesions were commonly observed in acinar-derived tumor sections. Immunofluorescence staining with human-specific antibody STEM121 and mCherry (expressed on the same cassette as oncogenic KRAS) confirmed the human origin of the cancer lesions (Figs. 3g, S3e). All of the tumor tissues had high levels of MUC5AC expression (Fig. 3f), which have been reported to express in metaplastic and cancer cells. We also observed expression of KRT19 and pERK in the tumors derived from both lineages (Fig. 3g). With these observations, we unambiguously demonstrated that both human acinar and ductal lineages are capable of transforming into malignant cells with a very similar phenotype. The striking similarities in histology and the expression of immunohistochemical markers between acinar- and ductal-derived tumors help us to understand why it is of great difficulty to identify the cellular origin of human PDAC in clinical samples.

## Transcriptional reprograming in acinar- and ductal-KPTS cells

A recent scRNA-seq analysis identified several cell populations from human PDAC samples, including acinar, duct-like, PanIN, and PDAC cells with unique gene expression patterns[24]. To compare our cells with the human PDAC samples, we examined the referenced cell type-specific gene expression in our RNA-seq data (Fig. 4a). Deconvolution analysis[21] was performed to infer the cell type composition in our models using treatment naïve human PDAC scRNA-seq samples[24] as a

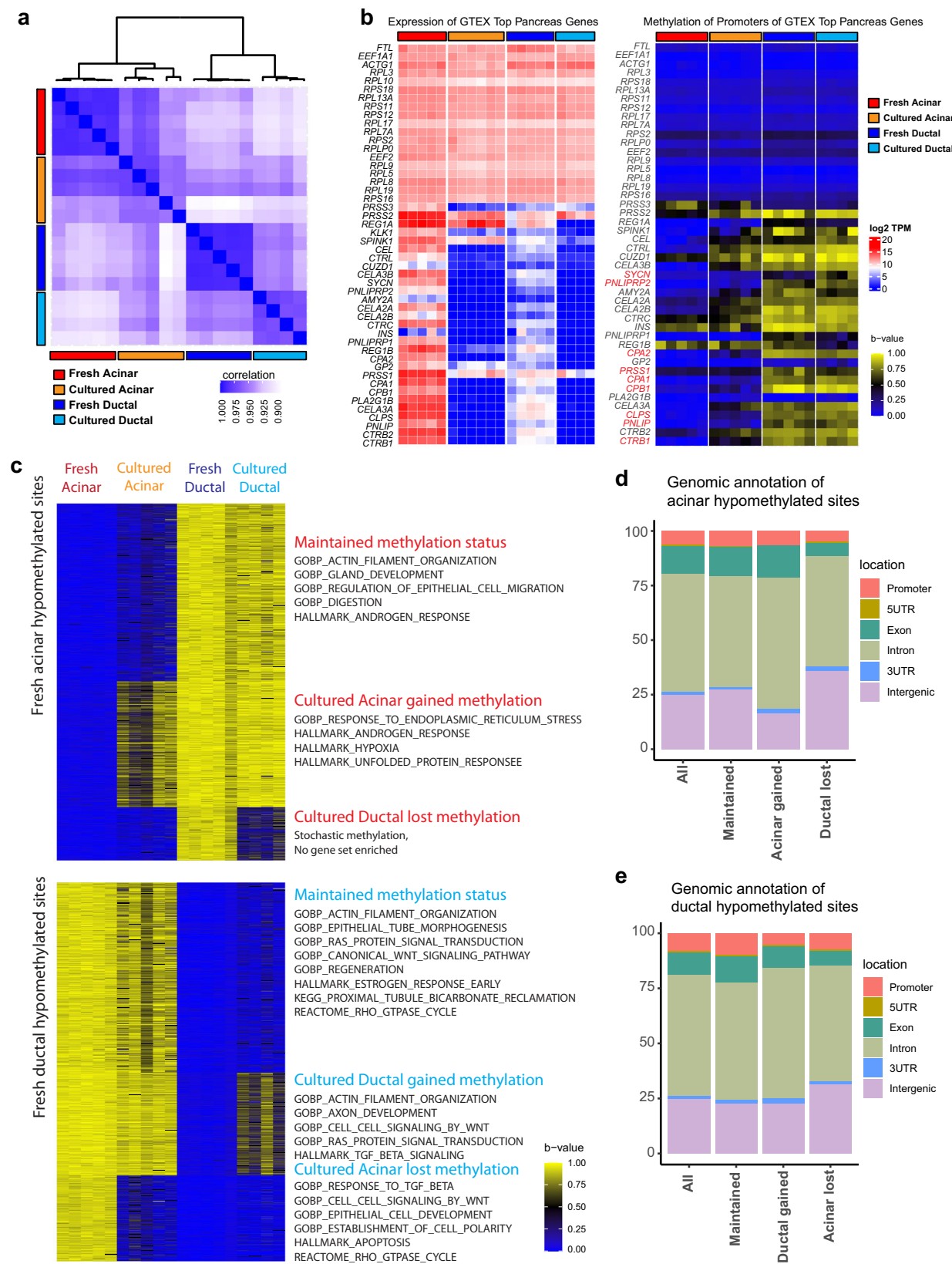

reference. The analysis inferred >99% of our fresh acinar cells as normal acinar cells, and ~80% of our fresh ductal cells as ductal-like 1 cells present in the referenced human PDAC samples, confirming the robustness of such analysis (Fig. 4b). Consistent to our previous observation that cultured acinar organoids acquired disease-associated metaplastic features, ~85% of our cultured metaplastic acinar organoids were inferred as duct-like 2 cells which were categorized as a major source of malignant PDAC cells in the referenced study (Fig. 4b). Interesting, nearly 100% of our acinar-KPTS cells were inferred as PDAC cell population present in the referenced treatment naive human PDAC cases, while our ductal-KPTS samples were inferred to contain a small fraction of PanIN and duct-like 1 cells (Fig. 4b). The

**Fig. 2 | Verification of cultured cell identity by DNA methylation status.**
**a** Correlation matrix of DNA methylation profiles of fresh acinar ($n = 5$), cultured acinar ($n = 5$), fresh ductal ($n = 5$) and cultured ductal cells ($n = 4$) by all surveyed methylation sites. **b** Expression heatmap (left) and DNA methylation status (right) of the promoters of the top genes expressed in the pancreas according to the GTEx database, in fresh and cultured cells of each lineage. **c** DNA methylation status in fresh and cultured cells of the mostly highly acinar hypomethylated sites (top, beta-value difference >0.8) and ductal hypomethylated sites (bottom). K means clustering identified 3 subgroups of sites with distinct methylation pattern across all groups of samples. Overrepresentation analysis for genes associated with each subgroup of sites was performed, with selected enriched gene sets annotated alongside their corresponding clusters. **d** Distribution of each subgroup of sites in (**c**, top) with reference to proximal genomic features. **e** Distribution of each subgroup of sites in (**c**, bottom) with reference to proximal genomic features.

similarity of our KPTS cells to the treatment naive human PDAC cells confirmed the clinical relevance of our model system.

To understand the early oncogenic reprograming in each lineage, we compared the acinar-KPTS cells to metaplastic acinar cells, and compared the ductal-KPTS cells to fresh ductal cells (Fig. S4a–c, Supplementary Data 5). Gene set overrepresentation analysis of the DEGs in KPTS cells of each lineage revealed diverse transcriptional changes in oncogenic cells, including common and lineage-specific changes (Fig. 4c, d). For example, cell cycle related gene sets were enriched in KPTS high genes from both lineages, indicating a common enhanced proliferation potential in oncogenic cells (Fig. 4c). KPTS cells from both lineages also shared enhanced expression of HALLMARK E2F target genes (Fig. 4e). Interestingly, gene sets related to telomere maintenance were only enriched in acinar-KPTS high genes, but not in ductal-KPTS high genes (Fig. 4c). A close look at the REACTOME extension of telomeres gene set revealed that many genes involved in the program, including *TERT*, were induced in acinar-KPTS cells, but not in ductal-KPTS cells (Fig. 4f). Other gene sets enriched in the KPTS high genes of both lineages include epithelial-mesenchymal transition (EMT), extracellular matrix organization, and RHO GTPase activate formins (Fig. S4d–f). The gene expression patterns across all the cell states revealed a dynamic transition from normal exocrine cells to an oncogenic cell state in response to environmental and genetic cues.

### Intrinsic alterations in immune signaling in acinar- and ductal-KPTS cells

The gene set analysis of DEGs revealed a downregulation of immune-related pathways in acinar-KPTS cells (Fig. 4d). A comprehensive comparison of genes involved in antigen processing and presentation revealed different expression patterns in the genes involved in different parts of the antigen presentation process (Fig. 5a). A group of Rab GTPases involved in endocytosis of extracellular antigens were induced in both lineages after culture (Fig. 5a, b). Genes encoding the Cathepsin family of proteases in the lysosome were only induced in cultured acinar cells but not ductal cells. In addition, gene expression of B2M and TAP1/2, the key components in the MHC I pathway, were also downregulated in acinar-KPTS cells (Fig. 5a, b). Interestingly, genes encoding several MHC-I molecules as well as PD-L1 (*CD274*) were found to downregulate in KPTS cells from acinar or both lineages (Fig. 5c). Consistently, by using an antibody of human HLA-A, B, C molecules, we confirmed the reduced MHC I expression in acinar-KPTS cells compared to paired metaplastic acinar organoids using flow cytometry analysis (Fig. 5d). Dynamic change of antigen presentation pathway without immune pressure suggests a possible intrinsic mechanism of immune regulation in epithelial cells during early PDAC tumorigenesis.

We also noted a general downregulation of genes involved in the complement cascade in both acinar and ductal-KPTS cells, including *C3*, *C4*, *C1R*, and *C1S* (Fig. 5e). However, the coagulation genes were upregulated in both acinar and ductal KPTS cells, which corroborates a previous finding that mice bearing pancreatic tumors have larger venous clots[25]. These intrinsic changes observed in early PDAC-like cells may shed light on the currently less understood implication of the coagulation and complement systems in pancreatic cancer tumorigenesis. Other altered pathways in the KPTS cells included enhanced glycolysis in acinar-KPTS cells and alterations in p38MAPK cascade, O-linked glycosylation, and TNFα signaling via NFkb in the KPTS cells

from both lineages (Fig. S5a–d). Overall, the analysis of common and lineage-specific transcriptional changes in KPTS cells revealed a proliferation-favored oncogenic phenotype with metabolic rewiring, ECM remodeling, EMT, and immune-evasion potential, resembling common cancer biology.

### Analysis of early human PDAC models identifies potential diagnostic biomarkers

As current translational studies in human PDAC have mainly focused on cancer cell lines and patient derived tumor samples, we took advantage of our unique early PDAC model to identify early diagnostic markers. To this end, we first identified 140 genes which were significantly upregulated in acinar-KPTS organoids when compared with normal and metaplastic cells (fold change >4, adj.$p < 0.05$) (Fig. 6a, Supplementary Data 6). Similarly, 696 genes were identified which were significantly upregulated in ductal-KPTS cells (Fig. 6b, Supplementary Data 6). We then curated 2802 genes which were significantly highly expressed in TCGA PDAC samples compared with the GTEx normal pancreas dataset (fold change >4, adj. $p < 0.05$). Comparison of all the three groups of genes identified 51 overlapped genes, including 26 genes encoding extracellular and membrane proteins, which hold potential as candidates for early diagnostic markers (Fig. 6c, d). It has been demonstrated that TCGA PDAC samples have substantial amount of stromal and immune composition[26]. Therefore, integration of our data from epithelial cells with the TCGA/GTEx comparison could help to minimize interference from mixed cell composition, while keeping the observation clinically relevant. Interestingly, among the 51 genes, *AREG*[27], *DKK1*[28], *EREG*[29], *NT5E*[30], *ZBED2*[31], *FAM83A*[32], and *S100A2*[33] are associated with PDAC prognosis and have been reported to play important roles in PDAC development. On the other hand, 27 of the 51 genes identified here either have not been associated with PDAC before or have unclear implications in PDAC development (e.g., *AHNAK2*, *SNCG*, *WNT10A*), thereby warranting further investigation.

To validate the clinical relevance of identified potential biomarkers (Fig. 6e), we first performed immunohistochemistry (IHC) staining of selected genes on our KPTS tumors and a human pancreatic cystic neoplasm as well as its adjacent normal tissue. Results showed that, all the tested proteins (AHNAK2, AREG, SEMA3A, SNCG and WNT10A) are highly expressed in our KPTS tumors originated from both lineages, as well as in the human pancreatic cystic tumor, but not in adjacent normal tissue (Figs. 6f, g, S6a). A more comprehensive validation was conducted by performing IHC staining on a human PDAC tissue array[34] containing paired tumor and adjacent normal area from 28 patients (Fig. 6h, i). Consistently, AHNAK2, a protein which was previously proposed as a PDAC prognosis marker with unknown function[35,36], was positive in tumor area of 25/28 patients, compared with only 1/28 patients positive at adjacent normal area (X-squared = 37.979, degree of freedom = 1, $p$ value = $7.1 \times 10^{-10}$). AREG, a member in EGF signaling which negatively correlates with PDAC prognosis[37], was positive in tumor area of 18/28 patients, compared with 9/28 patients positive at adjacent normal area (X-squared = 4.5773, degree of freedom = 1, $p$ value = 0.0324). Taken together, these data confirmed the high expression profile of tested genes in both our early PDAC-like cells as well as advanced clinical PDAC samples, demonstrating their potential as biomarkers for PDAC early detection, possibly when used as a gene panel.

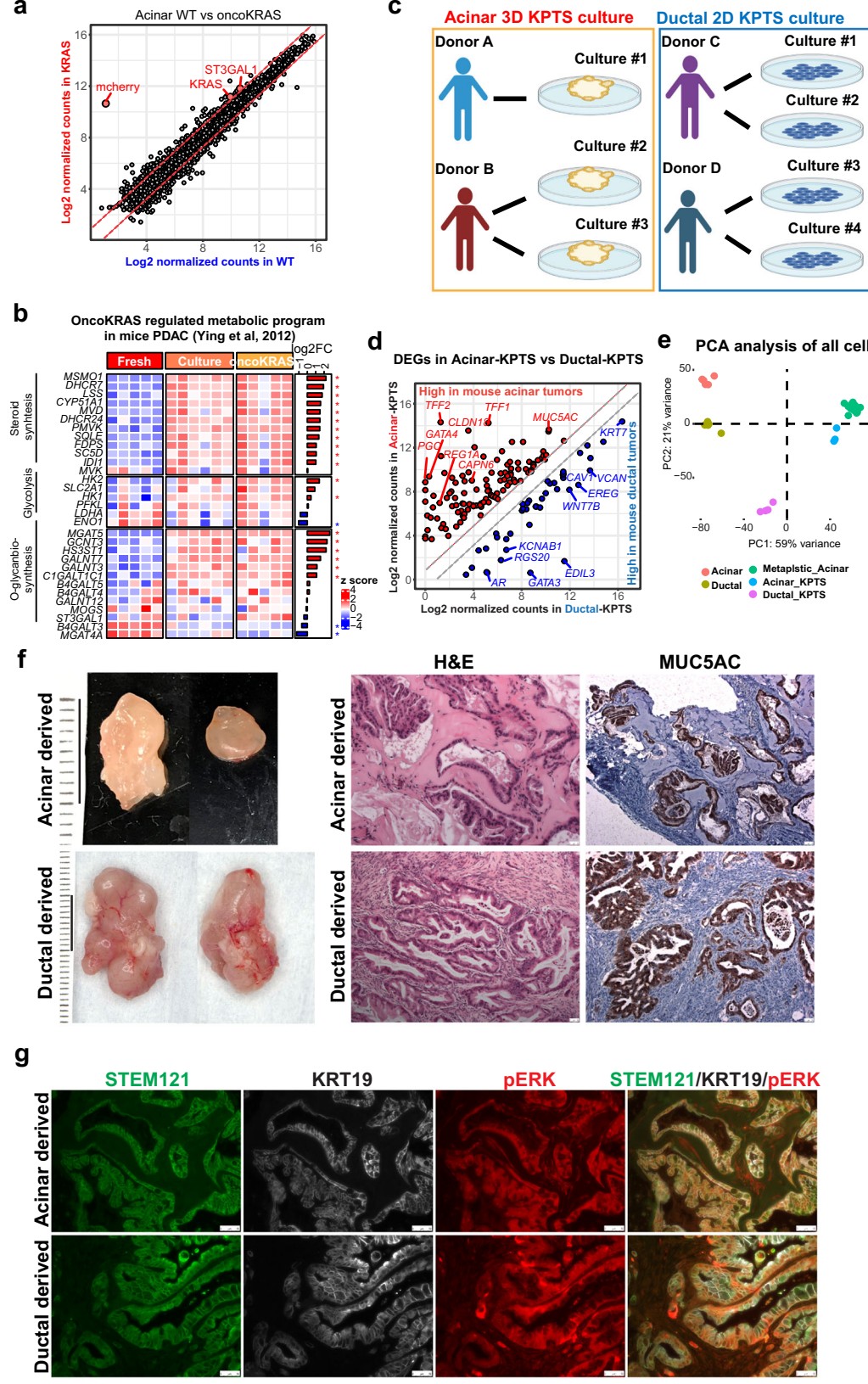

In addition to the 51 genes commonly found in all the three groups of genes, we also examined the 36 genes highly expressed in acinar-KPTS and TCGA PDAC samples, as well as 282 genes highly expressed in ductal-KPTS and TCGA PDAC samples (Fig. S6b). This led to identification of additional genes which may have lineage-specific implications in PDAC development.

## Discussion

Although the acinar origin of PDAC has been demonstrated in mouse models[8–13], determining the cellular origin of human PDAC has remained a conundrum. This is largely due to the difficulty of long-term culturing acinar cells, and the lack of a reliable method to verify cell lineage because of the rapid and profound alterations to

**Fig. 3 | Genetically engineering human primary acinar and ductal cells to form PDAC. a** Scatter plot of normalized gene expression in wild type and oncogenic *KRAS* expressing acinar organoids. The two annotated genes were the only significantly differentially expressed genes between two groups (fold change >2, adj.$p$ < 0.05, by negative binomial Wald test using DESeq2 1.36.0 in R software). *mCherry* was included in the oncogenic *KRAS* expressing vector. **b** Expression heatmap of listed genes in Acinar-derived fresh, cultured wild type, and cultured KRAS cells. The genes were curated in the indicated reference and were involved in *KRAS* mediated PDAC reprograming in mice. The side bar represents log2 fold change between cultured wild type vs fresh acinar cells (by negative binomial Wald test using DESeq2 in R software). Asterisk indicates adj.$p$ < 0.05. **c** Schematic demonstration of establishing acinar-KPTS 3D culture and ductal-KPTS 2D culture.

**d** Normalized expression of differentially expressed genes between acinar-KPTS and ductal-KPTS cells (fold change >2, adj.$p$ < 0.05, by negative binomial Wald test using DESeq2 in R software), which were also identified in mice acinar tumors versus ductal tumors. **e** PCA analysis of gene expression in all fresh, cultured and genetically modified acinar and ductal samples generated in the present work. **f** Left: representative photos of xenograft tumors derived from engineered acinar ($n$ = 15) and ductal ($n$ = 30) cultures. Right: H&E and MUC5AC IHC staining of xenograft tumors. Each staining was performed in at least 3 tumor sections with 8 fields of views for each. Scale bar: 50 μm. **g** Immunofluorescence staining of human specific STEM121, KRT19 and pERK of acinar-derived and ductal-derived tumor sections. The staining was performed in 6 independent tumor tissues with 3 sections for each. Scale bar: 50 μm.

transcription these cells undergo when stressed. While previous reports have considered their cultured pancreatic organoids to be ductal origin[7,38], these studies determined ductal identity by examining the expression of specific acinar and ductal markers, which we demonstrated could also be rapidly altered in cultured acinar cells. In comparison, here we sorted two lineages and verified lineage identity through comprehensive DNA methylation profiling, providing more authentic lineage-specific information. We observed that DNA methylation patterns were largely maintained in cultured human exocrine pancreatic cells, and could be used to ascertain the cell of origin. Consequently, we can conclude that previous organoid culture conditions support acinar lineage growth and we successfully generated organoids with long-term proliferation capacity from sorted primary human pancreatic acinar cells[39]. Genetically engineered acinar and ductal cells harboring four classic PDAC driver mutations were both capable of generating tumors in vivo. Therefore, we confidently conclude that human acinar and ductal cells can both develop into PDAC. It is of great interest to further investigate the degree to which the lineage specific methylation signatures are maintained in human PDACs.

We found that although cultured acinar organoids lost their most obvious lineage features and partially acquired ductal-like characteristics, they did not transdifferentiate into ductal cells. Instead, the acinar-derived organoids entered a unique state distinct from both normal acinar and ductal lineages. RNA expression of TFFs, mucins, as well as tuft cell markers known to be expressed by disease-specific populations identified in human pancreatitis patients were significantly up-regulated in acinar-derived organoids[20]. Our data provided experimental evidence to support the acinar cell origin of these disease-specific populations.

We noted that the wild-type acinar organoids, even without the presence of a *KRAS* mutation, already became metaplastic and acquired a disease-associated signature. This observation might be explained by the facts that the acinar cells are highly plastic and prone to phenotypic change under in vitro culture stress, and that the partial activation of oncogenic *KRAS* pathway can be induced from the supplementation of EGF in the growth media. This aligns with findings from mouse models where a proto-oncogenic transcriptional program is initiated by pancreatic injuries before the acquisition of *KRAS* mutations[16,17]. Thus, introducing oncogenic *KRAS* to acinar culture did not cause significant phenotypic changes. We postulated that oncogenic *KRAS* in acinar cells may be an essential prerequisite, rather than a sole driver, for PDAC transformation in our model system.

Our model recapitulated the disease progression-associated transcriptional transition, highlighting the unique advantage of our culture model in investigating the mechanistic aspects of early human PDAC tumorigenesis. The proliferation potential showed a progressive enhancement from normal cells to early PDAC-like cells. The upregulation of coagulation genes in both acinar and ductal KPTS cells provides a possible mechanism for increased venous clots in PDAC mouse model. Further studies are needed to illustrate the roles of the coagulation and complement cascade in PDAC tumorigenesis. MHC

molecules were down-regulated in KPTS organoids after acquiring *p16*, *TP53* and *SMAD4* mutations especially in acinar lineage. Recent studies reported that *TP53* increases the expression of MHC molecules via *ERAP1*[40] and *IRF2*[41], whose expression remained unchanged in our organoids, suggesting that MHC molecules were down-regulated by other mechanisms. In addition, the down-regulation of MHC molecules in KPTS organoids and PD-L1 in acinar lineage occurred in the absence of selective pressure from an active immune system, indicating the intrinsic effects of cell lineage and the oncogenic mutations on promoting immune escape in PDAC cells.

One of the major obstacles to improving human PDAC survival is the lack of early diagnostic methods. Current human PDAC studies mainly rely on cancer cell lines or clinical samples derived from PDAC patients who already have symptomatic disease, which are not suitable for identifying early diagnostic markers. In comparison, our engineered cells acquired both PanIN histology and a PDAC associated transcriptomic signature, suggesting an early PDAC-like state, which is in a unique position to investigate early tumorigenesis and identify early diagnostic markers. In the present study, we took advantage of our early PDAC model and identified a group of genes with elevated expression that align with early PDAC progression, which were also found to be highly expressed in TCGA PDAC samples compared with the GTEx normal pancreas dataset. Many of these genes encode extracellular and membrane proteins, which hold potential as PDAC diagnostic markers. Using clinical human PDAC samples, we validated the protein expression patterns of selected genes in tumor and adjacent normal area, including *AHNAK2*, *AREG*, *SEMA3A*, *SNCG*, and *WNT10A*. Particularly, *AHNAK2* was previously proposed as a PDAC prognosis biomarker[42], corroborating the robustness of our findings. Interestingly, many of the genes identified in this study have either not been previously associated with PDAC or have unclear implications in PDAC development, warranting further investigation.

One limitation of our study is that the acinar and ductal cultures were established in different settings, i.e., 3D organoids vs 2D monolayers. It had been previously reported that the stiffness of tumor environment is associated with cancer progression[43]. The average stiffness of human pancreatic PDX tissues is 1000–2000 Pa (the Young's modulus), with some specific areas reaching a stiffness of up to 20,000 Pa[43]. In comparison, Matrigel, similar to the matrix used in this study for 3D cell culture, is much softer with a stiffness of only ~90 Pa. Indeed, De la Peña et al. revealed that different culture conditions resulted in differences in cell transcriptomic profile, interaction with stromal components, as well as drug response[43]. Thus, the different features of acinar- and ductal-KPTS cells and tumors could be attributed to the different stiffness in 3D vs 2D culture environment. On the other hand, the fact that acinar cells can be established as 3D organoids and ductal cells can only be established in 2D culture with PDAC-mutations suggests a difference in stiffness requirements of each lineage during PDAC initiation. In addition, we observed elevated *TERT* expression in acinar lineage but not in ductal lineage, which may contribute to limited growth of ductal cells. Using lineage tracing of

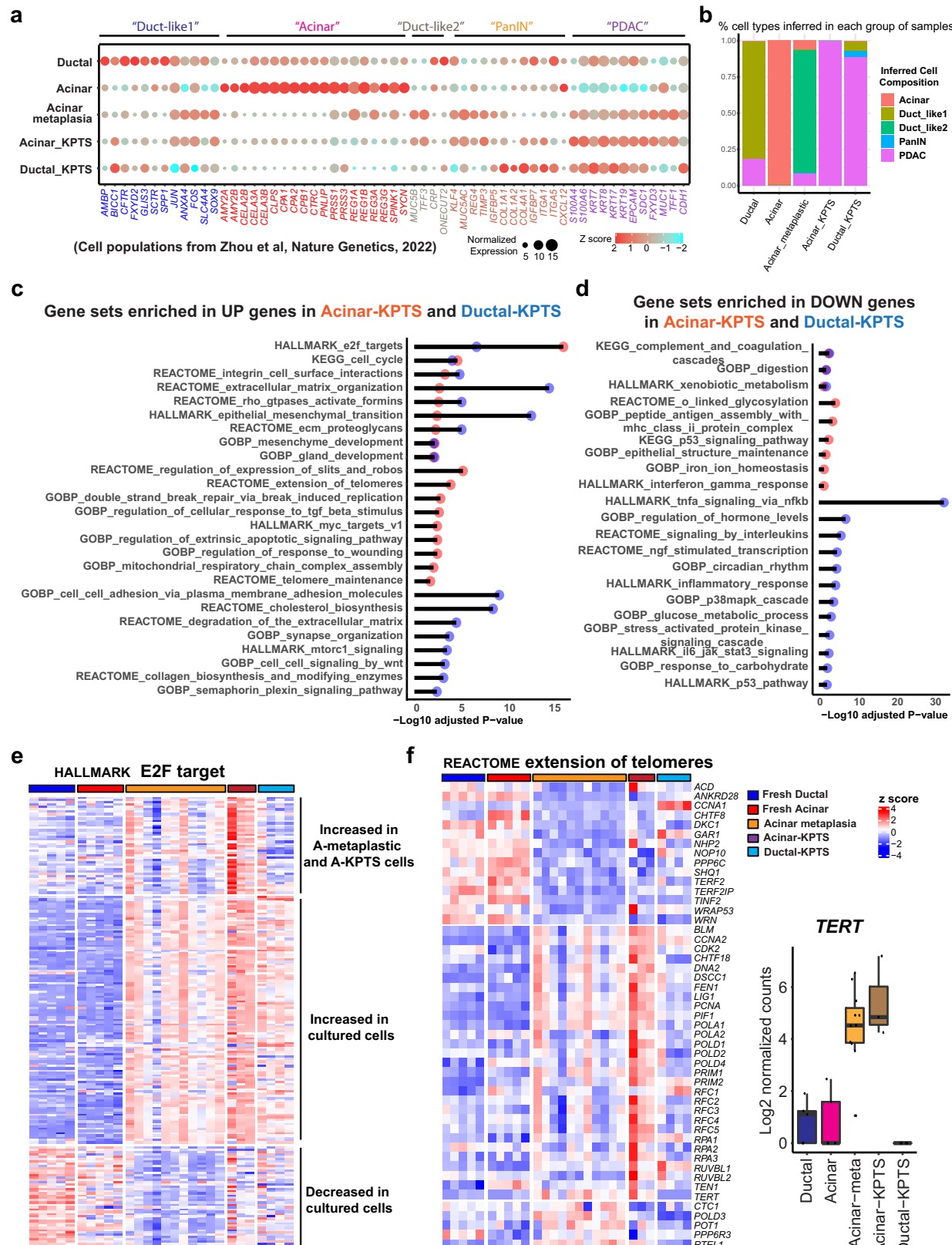

endogenous *Tert* locus, Neuhöfer et al., have identified a rare TERT-positive subpopulation of pancreatic acinar cells dispersed throughout the mouse exocrine compartment[44]. The higher expression of *TERT* in our acinar metaplastic cells could be upregulation of *TERT* or expansion of existing *TERT* high cells, similarly to what was observed in the mouse model. Further study is required to understand the

differences between acinar and ductal cells in tumorigenic capacity, interaction with tumor environment, and drug response.

Considering the current lack of an early human PDAC model, engineering normal human primary pancreatic cells offers an opportunity to decipher the initiation and early progression of human PDAC. This is a crucial step toward the goal of developing prevention and

**Fig. 4 | Transcriptomic reprograming in oncogenic cells derived from acinar and ductal lineages. a** Expression of indicated cell type signature genes (columns, identified from indicated reference) in our fresh, metaplastic, and KPTS cells (rows). The dot size is proportional to the normalized read counts, the color represents z score of gene expression. **b** Bar plot of inferred cell type composition in our samples using treatment naive human PDAC scRNA seq samples (described in the indicated work as shown in **a**) as a reference. The bulk RNA sample deconvolution was performed by using MuSiC method. **c** Overrepresentation analysis for upregulated genes in Acinar-KPTS cells (vs Acinar-metaplastic cells) and in Ductal-KPTS cells (vs fresh ductal cells). Overrepresentation analysis was performed using the clusterProfiler 4.4.4 package in R software with default settings. A significant enrichment was considered with multiple-test adjusted *p* value < 0.05.

**d** Overrepresentation analysis for downregulated genes in Acinar-KPTS cells (vs Acinar-metaplastic cells) and in Ductal-KPTS cells (vs fresh ductal cells). The analysis was performed using the clusterProfiler 4.4.4 package. A significant enrichment was considered with multiple-test adjusted *p* value < 0.05. **e** Expression heatmap of HALLMARK E2F target genes in fresh, metaplastic, and KPTS cells. K means clustering was performed to identify different expression patterns across all groups of samples. **f** Left: Expression heatmap of genes in the REACTOME extension of telomeres gene set in fresh, metaplastic, and KPTS cells. Right: Boxplot of *TERT* expression in all groups of samples. The middle line of box represents the median value, the bounds of box represent the IQR, and the whiskers extend to $1.5 \times$ IQR. Fresh ductal $n = 5$, fresh acinar $n = 5$, acinar-metaplastic $n = 11$, acinar-KPTS $n = 3$, ductal-KPTS $n = 4$.

early diagnosis strategies for improving outcomes of this disease. Even though we present this unique model for studying cancer initiation from human primary pancreatic cells, where obtaining the human tissues from the early stages of pancreatic cancer is nearly impossible, the question remains as to how well the in vitro disease model can recapitulate the malignant process in patients. It would be interesting to use single-cell RNA sequencing on cultured organoids and xenograft tumors to experimentally compare our model with clinical samples to further validate the clinical relevance. In addition, adding immune components with humanized mice in our model may provide a better tumor microenvironment, making it more relevant to clinical settings.

In conclusion, we provided comprehensive experimental evidence that both human pancreatic acinar and ductal cells are capable of transforming into PDAC. We reconstructed the exocrine cell transformation process and revealed a dynamic change of transcriptomic profiles from a normal cell state to the oncogenic state. We reported a list of genes which were highly expressed during early transformation and in clinical PDAC samples which may represent candidates for PDAC diagnostic tests. These observations provide additional insights into the lineage specific aspects of human PDAC pathogenesis, and may assist future endeavors in the improvement of disease diagnosis, prevention, and treatment.

## Methods

### Ethical statement
All experiments involving normal human primary pancreatic tissues from organ donors were reviewed by the institutional review board at UT Health San Antonio. The tissues were de-identified, with only information on sex, race, age, weight, height and cause of death. The IRB committee has agreed that the project does not require IRB approval because it is either: Not human research as defined by DHHS regulations at 45 CFR 46 and FDA regulations at 21 CFR 56; The project does not include non-routine intervention or interaction with a living individual for the primary purpose of obtaining data regarding the effect of the intervention or interaction, nor do the researchers obtain private, identifiable information about living individuals.

Human pancreatic cystic neoplasm sample was collected and prepared at UT Health at San Antonio with patient's consent in accordance with the guidelines. Experiment involving this clinical human PDAC sample was approved by the institutional review board at UT Health San Antonio (IRB protocol #20230412NHR).

All animal experiments were approved by the Institutional Animal Care and Use Committees at UT Health San Antonio (protocol #20130023AR), and were performed in accordance with relevant guidelines and regulations.

### Isolation of human pancreatic cells from organ donors using flow cytometry
Human islet-depleted cell fractions were obtained from organ donors deceased due to acute traumatic or anoxic death by Prodo Laboratories, Inc, and were shipped overnight to our laboratory ($n = 21$, see donor information in Supplementary Data 1). The flow sorting

procedures were performed as previously described with some modifications[14]. Briefly, exocrine tissue cells were incubated with FITC-conjugated UEA-1 (0.25 μg/ml, Vector Laboratories, Newark, CA, FL-1061-5) for 10 min. at 4 °C and washed with PBS. After washing, the cells were digested with TrypLE™ Express (Life Technologies, Grand Island, NY, 12605-028) for 5–8 min at 37 °C. Cells were collected by centrifugation and washed with FACS buffer (10 mM EGTA, 2% FBS in PBS). After washing, cells were stained with Pacific blue-conjugated anti-CLA (BioLegend, San Diego, CA, 321308) and anti-7AAD (BioLegend, San Diego, CA, 420404) for 15 min at 4 °C. Cell pellets were collected by centrifugation and washed with PBS after staining. The cells were sorted using a FACSAria™ II (BD Biosciences, San Diego, CA) and collected in 100% FBS. After sorting, cells were washed with serum-free Advanced DMEM/F-12 media (Life Technologies, Grand Island, NY, 12634-010).

### 3D organoid culture of sorted acinar and ductal cells
Sorted primary human acinar and ductal cells were cultured as 3D organoids[7] embedded in the Type 2 Cultrex RGF Basement Membrane Extract (BME, R&D Systems, Minneapolis, MN, 3533-010-02 P). Briefly, ~0.5 × 10⁶ freshly sorted cells were washed with PBS and centrifuged. The cell pellet was resuspended in 10 μL ice cold BME, and placed in the bottom of a pre-warmed 24 well plate. After solidification at 37 °C for 5–10 min, 500 μL of organoid growth media were added into each well. The organoid growth media was composed of serum free advanced DMEM/F-12 media (Life Technologies, Grand Island, NY, 12634-010) supplemented with Wnt3a conditioned medium (50%)[45], recombinant human R-Spondin 1 (500 ng/ml, R&D Systems, Minneapolis, MN, 4645-RS), Noggin (200 ng/ml, R&D Systems, Minneapolis, MN, 6057-NG), FGF10 (100 ng/ml, R&D Systems, Minneapolis, MN, 345-FG), EGF (50 ng/ml, R&D Systems, Minneapolis, MN, 236-EG), PGE II (1 nM, Fisher Scientific, Waltham, 22-961-0), A83-01 (0.5 μM, R&D Systems, Minneapolis, MN, 2939), Nicotinamide (10 mM, Thermo Fisher Scientific, Waltham, MA, 48-190-7100GM), Penicillin/Streptomycin (1 mM, Sigma-Aldrich, St. Louis, MO, P4333), HEPES (10 mM, Life Technologies, Grand Island, NY, 15630), GlutaMax™ (1X, Life Technologies, Grand Island, NY, 35050). The growth media was refreshed every 3-4 days until cells reached confluency. For subculture, the organoids were collected and digested with TrypLE™ Express (Thermo Fisher Scientific, 12605-028), followed by gentle pipetting to release cells from extracellular matrix. Finally, a small fraction of cells were embedded in fresh BME for culture.

### 2D monolayer culture of sorted ductal cells
As ductal cells could not survive in 3D culture for more than 3–4 passages, we established long term 2D ductal culture for further experiments. Briefly, the floor of a 24 well plate was coated with 100 μL of 5% BME in advanced DMEM/F-12 followed by incubation at 37 °C for 5–10 min. The BME solution was then removed and replaced with a suspension of ~0.5 × 10⁶ freshly sorted ductal cells in 500 μL of advanced DMEM/F-12 media supplemented with 5% serum. The media was refreshed every 3–4 days until cells reached confluency, at which

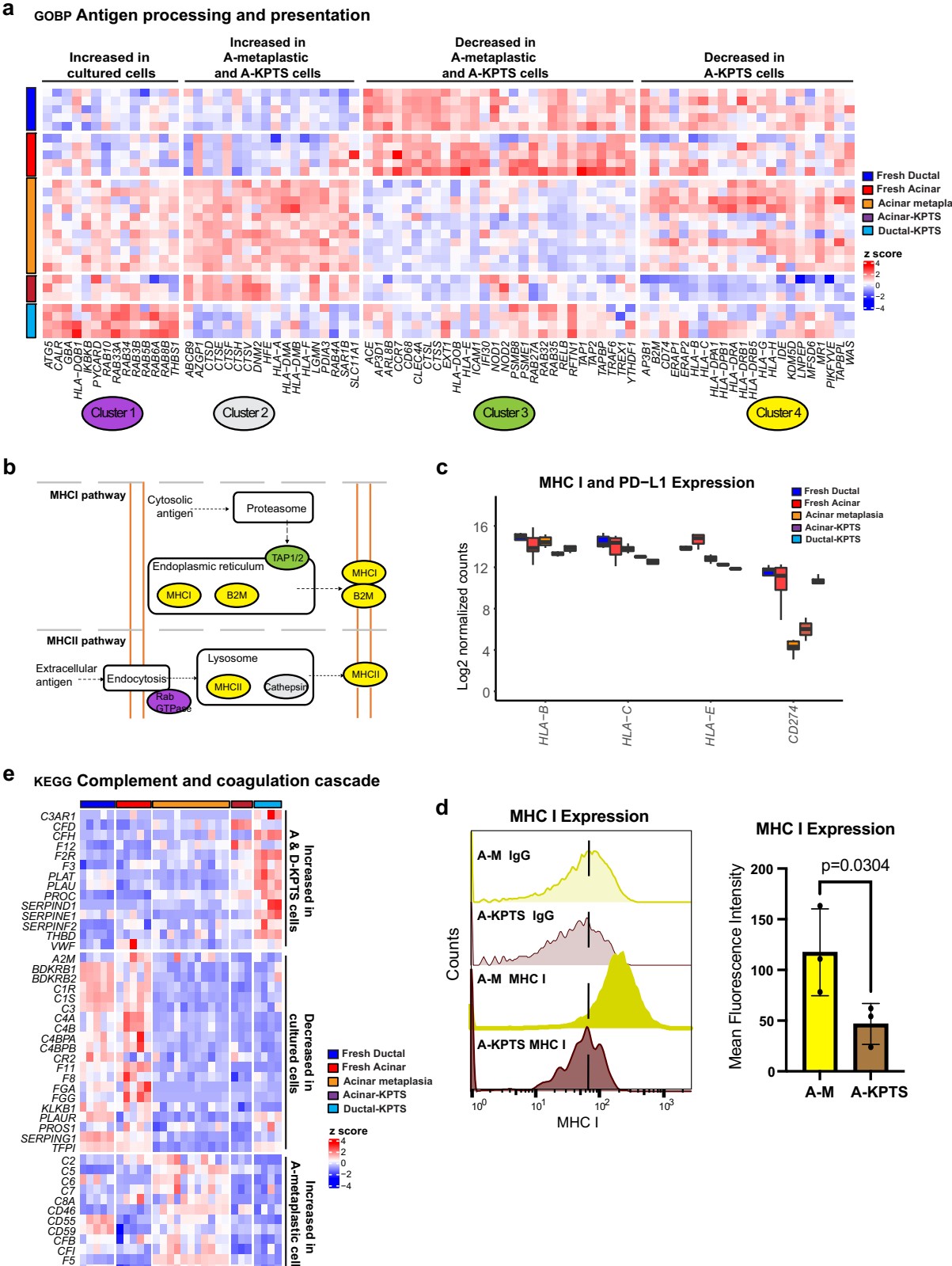

point TrypLE™ Express was used to digest the cells for passage or for further experiments.

### Electron microscopy

Transmission Electron Microscopy (TEM) was performed at the UT Health San Antonio Electron Microscopy Laboratory. The basic processing steps were the same as previously described[46]. The freshly sorted acinar and ductal cells were fixed with a solution containing 1% glutaraldehyde plus 4% paraformaldehyde for 30 min. Tissue was imbedded in resin, then cut into 90 nm slices, and placed on copper grids. Uranyl acetate was used to stain the grids for 30 s in the microwave, followed by 20 s of Reynolds' lead. AMT (advanced

**Fig. 5 | Intrinsic alteration of immune program in oncogenic acinar and ductal cells. a** Expression heatmap of genes in the GOBP antigen processing and presentation gene set in fresh, metaplastic, and KPTS cells. K means clustering was performed to identify different expression patterns across all groups of samples. **b** Schematic illustration of key gene changes in antigen processing and presentation pathways in cultured acinar and ductal cells. **c** Boxplot of RNA expression of MHC class I and PD-L1 (*CD274*) genes in all groups of samples. The middle line of box represents the median value, the bounds of box represent the IQR, and the whiskers extend to 1.5 × IQR. Fresh ductal $n = 5$, fresh acinar $n = 5$, acinar-

metaplastic $n = 11$, acinar-KPTS $n = 3$, ductal-KPTS $n = 4$. **d** Left: representative flow cytometry analysis of MHC I molecule and mouse IgG isotype control from 3 independent paired metaplastic and KPTS acinar organoids. Right: quantification of MHC I mean fluorescence from 3 independent paired metaplastic and KPTS samples. The fluorescence intensity of MHC I in each sample was adjusted by subtracting the intensity of corresponding isotype control, before analyzed by two-tailed Student's *t* test. Error bar represents standard deviation. **e** Expression heatmap of genes in the KEGG complement and coagulation cascade gene set in fresh, metaplastic, and KPTS cells.

microscopy methods) software was used to image samples at 80 kV using a JEOL 1230 electron microscope.

## RNA-seq sample preparation
Cell pellets of freshly sorted and cultured cells were collected for total RNA extraction using Zymo Direct-zol RNA Miniprep Kit (Zymo Research, Irvine, CA, R2063) following manufacturer's instructions. RNA quality was verified by 2100 Bioanalyzer (Agilent Technologies) and Qubit 4 Fluorometer (Thermo Fisher Scientific). Indexed cDNA libraries were prepared from high quality RNA samples using the Illumina® Stranded mRNA Prep Ligation kit (Illumina, San Diego, CA, 20040532) following manufacturer's instructions. The cDNA libraries were then submitted to the Greehey Children's Cancer Research Institute (GCCRI) Genome Sequencing Facility at UT Health San Antonio for high throughput sequencing analysis using Illumina HiSeq 3000 or NovaSeq 6000 System.

## RNA-seq data analysis
Sequencing reads were aligned to reference genome GRCh38 using TopHat 2.1.1 and gene expression was quantified using HTSeq 0.11.1. Differential expression analysis of the read counts from aligned RNA seq data was performed using DESeq2 1.36.0 package in R software. A pre-filtering process was applied for each analysis to remove genes with low read count, i.e., only keep genes with >10 reads in at least half of the samples of at least one comparing group. For two-group comparison, the DEGs were defined as fold change >2, and Benjamin-Hochberg (BH) adjusted *p*-value < 0.05. For three-group comparison, the DEGs were identified using the LRT function from DESeq2 1.36.0 package, and defined as BH adjusted *p* value < 0.05. K Means cluster was performed using the Complexheatmap 2.13.2 package in R in order to identify gene clusters with certain expression pattern. Gene expression heatmaps were generated using TPM (transcript per million) or z-score of transformed/normalized read count as indicated in each figure. Overrepresentation analysis for lists of genes were performed using the clusterProfiler 4.4.4 package in R with default settings. Gene Set Enrichment Analysis (GSEA) for comparing two groups of expression dataset was performed using GSEA software downloaded from the BROAD Institute. A significant enrichment was considered with multiple-test adjusted *p* value < 0.05. For PCA analysis, all the bulk RNA-seq data generated from our work were normalized followed by PCA analysis using DESeq2 package.

## Comparison of bulk RNA-seq data with reference scRNA-seq data
Raw read count and metadata of referenced scRNA-seq data were downloaded according to the authors' instructions from corresponding references[15,19,20,24]. For PCA analysis, a pseudo-bulk RNA-seq data was created for each cell population present in referenced scRNA-seq data by aggregating the read count of all the cells present in each population using Seurat 4.3.0.1 package in R software. Then all the pseudo-bulk RNA-seq data as well as the RNA-seq data generated from our work were normalized using DESeq2 1.36.0 package and corrected for batch effect using limma 3.52.4 package in R software, followed by PCA analysis using DESeq2. Deconvolution of our bulk RNA-seq data

was performed using MuSiC 1.0.0 package in R software with previously characterized scRNA-seq data as references[19,24]. When using human PDAC scRNA-seq samples as a reference, only treatment naïve human PDAC samples were included. For most accurate deconvolution analysis, the referenced scRNA-seq data was pre-processed in Seurat 4.3.0.1 and trimmed to only contain top DEGs in each cell type present in the dataset.

## DNA methylation sequencing
Genomic DNA was extracted from pellets of freshly sorted and cultured pancreatic acinar and ductal cells using the Zymo Quick-DNA™ MiniPrep kit (Zymo Research, Irvine, CA, D3024) as according to manufacturer instructions. DNA quality was verified by Qubit 4 Fluorometer (Thermo Fisher Scientific). Library preparation was performed on 500–1000 ng of genomic DNA using the TruSeq Methyl Capture EPIC Library Prep Kit (Illumina, San Diego, CA, FC-151-1003), with which bisulfite conversion and PCR amplification of regions of interest was performed. Sequencing of the resulting reduced-representation DNA methylation sequencing libraries was performed using an Illumina NovaSeq 6000 at the GCCRI Genome Sequencing Facility. Alignment of sequencing files to GRCh38 and CpG calling was performed using Bismark 0.19.1. For subsequent analysis, only CpG sites for which there were at least 10 aligned reads for all samples were used. Methylation at a CpG site was calculated as the number of methylated reads divided by the total number of reads at that site (β-value), and the methylation of a genomic region was calculated as the mean of the β-values of all sites within that region. Differential methylation between regions was calculated using two-tailed Student's *t* test, with a region being considered differentially methylated if the FDR corrected *p* value was less than 0.05.

## Organoid histology sample preparation
The cultured organoids were collected by pipetting, washed with PBS twice, and fixed in 10% formalin for 4 h. The organoids were washed with PBS again, then resuspended in 50 µl of pre-warmed 4% low-melting point agarose (48 °C). Then the cell suspension was quickly transferred onto a cold glass slide on ice, which was maintained on ice for an additional 5 min to allow solidification. The solid agarose was transferred to a tissue block cassette and kept in 70% ethanol for further paraffin block preparation and histological examination.

## Genetic engineering of human primary pancreatic cells
A plasmid expressing oncogenic KRAS^G12V cDNA and mCherry was a generous gift from Seung Kim (Stanford University)[39]. CRISPR guide RNAs for *p16*, *TP53*, and *SMAD4* were designed and cloned into lentiCRISPR v2 plasmids (Addgene, Watertown, MA, #52961). Lentiviruses containing these expression constructs were packaged in 293 T cells by co-transfection with packaging plasmids pMD2.G (Addgene, Watertown, MA, #12259) and psPAX2 (Addgene, Watertown, MA, #12260). The sorted primary acinar and ductal cells were transduced with lentivirus with supplement of polybrene. The transduced cells were then subject to antibiotic and/or functional selections. For selection of KRAS^G12V transduction, G418 (1000 ug/mL) was added into organoid culture medium. For selection of CRISPR-*p16* transduction,

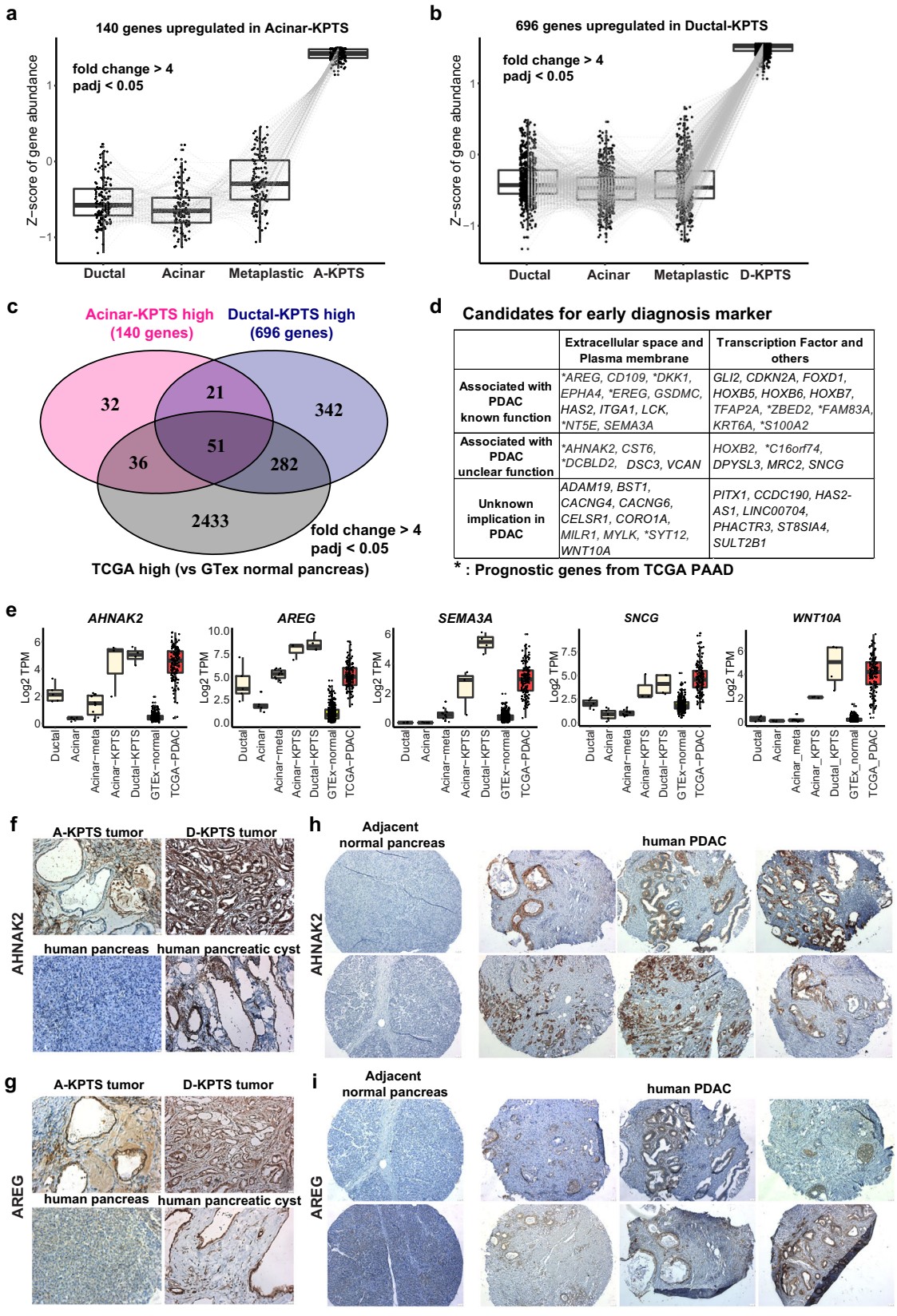

puromycin (1 µg/ml) was added into the culture medium. For selection of *TP53* mutation, 10 µM nutlin-3 was added. For selection of *SMAD4* mutation, Noggin-free media was used for cell culture, supplemented with 200 ng/mL BMP4. The expression of *mCherry* in the *KRAS^{G12V}* cassette was confirmed by RNA seq analysis. The mutations in *p16*, *TP53*, and *SMAD4* were confirmed by Sanger sequencing.

Oligo DNA sequences:
*p16* sgRNA: GGCTGGCCACGGCCGCGGCC
*TP53* sgRNA: ACTTCCTGAAAACAACGTTC
*SMAD4* sgRNA: TACGAACGAGTTGTATCACC
*p16* mutation genotyping forward primer: CGGTCCCTCCAG AGGATTTG

**Fig. 6 | Lineage-specific early PDAC model reveals candidates for diagnostic markers. a** Expression of 140 genes which were significantly overexpressed in acinar-KPTS cells compared with fresh ductal, fresh acinar, and acinar metaplastic cells (fold change >4, adj. $p < 0.05$, by negative binomial Wald test using DESeq2 1.36.0 in R software). The middle line of box represents the median value, the bounds of box represent the IQR, and the whiskers extend to $1.5 \times$ IQR. **b** Expression of 696 genes which were significantly overexpressed in ductal-KPTS cells compared with fresh ductal, fresh acinar, and acinar metaplastic cells (fold change >4, adj. $p < 0.05$, by negative binomial Wald test using DESeq2). The middle line of box represents the median value, the bounds of box represent the IQR, and the whiskers extend to $1.5 \times$ IQR. **c** Venn diagram of the intersections of 140 acinar-KPTS high genes, 696 ductal-KPTS high genes, and 2805 TCGA high genes. The 2805 TCGA high genes were identified by comparing TCGA PDAC samples with the GTEx normal pancreas dataset (fold change >4, adj. $p < 0.05$, by negative binomial Wald test using DESeq2). **d** Annotation of 51 overlapped genes identified in (**c**).

**e** Expression of selected genes in our samples (fresh ductal $n = 5$, fresh acinar $n = 5$, acinar-metaplastic $n = 11$, acinar-KPTS $n = 3$, ductal-KPTS $n = 4$) as well as GTEx pancreas ($n = 328$) and TCGA PDAC samples ($n = 149$). The middle line of box represents the median value, the bounds of box represent the IQR, and the whiskers extend to $1.5 \times$ IQR. **f–g** Immunohistochemistry staining of AHNAK2 (**f**) and AREG (**g**) proteins in the KPTS tumors generated in this work ($n = 2$ with 8 field of views for each), and a human pancreatic cystic neoplasm sample as well as adjacent normal pancreas (8 field of views for each). Scale bar: 25 μm. **h–i** Representative immunohistochemistry staining of AHNAK2 (**h**) and AREG (**i**) proteins in a human PDAC array including both tumor and adjacent normal pancreas from 28 PDAC patients (2 tumor tissues and 1 adjacent normal tissue per patient). Scale bar: 50 μm. Pearson's Chi-squared test with Yates' continuity correction was used to assess the statistical significance of the number of positive staining between tumor tissues versus normal tissues.

*p16* mutation genotyping reverse primer: TGGAGGCTAAGTAG TCCCAG
*TP53* mutation genotyping forward primer: TGCTGGATCCCC ACTTTTCC
*TP53* mutation genotyping reverse primer: GGATACGGCCAGGC ATTGAA
*SMAD4* mutation genotyping forward primer: CTGAGCACAGGCC TTGAAATTA
*SMAD4* mutation genotyping reverse primer: GTCGCGGGCTAT CTTCCAAA

## Flow cytometry analysis of MHC molecule expression

Paired wild type acinar organoids and acinar-KPTS cells were collected by TrypLE™ Express digestion, and washed with PBS. After washing, cells were stained with FITC anti-human HLA-A,B,C antibody (BioLegend, San Diego, CA, 311403), FITC anti-human HLA-DR, DP, DQ antibody (BioLegend, San Diego, CA, 361705), and FITC Mouse IgG2a isotype control (BioLegend, San Diego, CA, 400209) for 30 min at 4 °C. Cell pellets were collected by centrifugation and washed with FACS buffer after staining, followed by flow cytometry analysis using a BD LSRII (BD Biosciences). Data was analyzed with Flowjo v10 software.

## Animal experiments

All animal experiments were approved by the Institutional Animal Care and Use Committees at UT Health San Antonio (20130023AR), and were performed in accordance with relevant guidelines and regulations. The maximal tumor size in each mouse did not exceed 2 cm at the largest diameter as permitted by institutional guidelines. Four to six-week old female and male NSG (NOD.Cg-*Prkdc^scid^Il2rg^tm1Wjl^*/SzJ) mice (Strain #:005557) were purchased from The Jackson Laboratory. The mice were housed at a maximum of five per cage in a pathogen-free system with water and food ad libitum, with 12 h day/night cycle at 20–25 °C and 50–60% humidity. For subcutaneous transplantation, ~$1 \times 10^6$ cells were collected and suspended in 100 μL of 50% BME/50% serum-free Advanced DMEM/F-12 on ice. Tumor xenografts were established by subcutaneously injecting the cell mixture into the hind flank of NSG mice. The mice were maintained with a standard diet for 3 months to allow tumors to grow. At end point, the mice were euthanized, and the tumor tissues were collected for further histological analysis. The animal sex was not considered in the study design and analysis as no evidence for the impact of sex on the subcutaneous tumorigenesis of pancreatic cancer cells.

## Immunofluorescence and IHC staining of tissue sections

Human pancreatic cystic neoplasm sample was collected from a 71-year old black male at UT Health at San Antonio with patient's consent in accordance with the guidelines (IRB protocol #20230412NHR). Human PDAC tissue array contained tumors and adjacent normal area were previously generated from patients who underwent pancreatectomy at MD Anderson Cancer Center from 1990 to 2010[34]. For Immunostaining, paraffin-embedded organoid/tissue block were deparaffinized, rehydrated, and submerged in 200 °C heated R-Universal Epitope Recovery Buffer solution (Electron Microscopy Sciences, Hatfield, PA, #AP0530) for 30 min and then let cool at room temperature for 25 min. Sections were permeabilized using 0.5% PBST (0.5% Triton X-100, Acros Organics, Fair Lawn, NJ) for 10 min. Sections were subsequently blocked with 5% donkey serum in 0.1% PBST for 35 min at room temperature. Sections were then incubated with primary antibodies diluted in 5% donkey serum in 0.1% PBST at 4 °C overnight. For immunofluorescent staining, sections were incubated with fluorescent-tagged Alexa Fluor secondary antibodies (1:250, Jackson ImmunoResearch, West Grove, PA) diluted in 5% donkey serum in 0.1% PBST for 1 h at room temperature. Additionally, sections were incubated with DAPI (1:1000, Invitrogen, Carlsbad, CA, P36935) for 4 min at room temperature. Finally, sections were covered with a drop of VectaShield Vibrance Antifade Mounting Medium (Vector Laboratories, Inc., Burlingame, CA, H-1700). For immunohistochemistry, following primary antibody incubation, sections were incubated with biotinylated secondary antibody diluted in 5% donkey serum in 0.1% PBST for 1 h at room temperature. Then, sections were incubated with Streptavidin-Horseradish Peroxidase Pre-diluted (1X, BD Pharmingen, San Diego, CA, 51-75477E) for 1 h at room temperature. The color-complex was developed using DAB Substrate Kit (BD Pharmingen, San Diego, CA, 550880). Finally, sections were dehydrated and covered with a drop of Poly-Mount (Polysciences, Inc., Warrington, PA, 08381-120). All images were captured using Leica DMI6000 B microscope and software (Leica Microsystems, Buffalo Grove, IL). All primary and secondary antibodies used are listed below:

Mouse anti human STEM121 (Takara Bio, #Y40410), dilution 1:50;
Mouse anti human/mouse/rat KRT19 (DSHB, #Troma-III), dilution 1:50;
Rabbit anti human Alpha-Amylase (Sigma Aldrich, #A8273), dilution 1:500;
Mouse anti human Mucin5AC (Santa Cruz Biotechnology, #sc-33667), dilution 1:50;
Rabbit anti human/mouse WNT10A (Thermo Fisher, #26238-1-AP), dilution 1:50;
Rabbit anti human AHNAK2 (Sigma Aldrich, #HPA002940), dilution 1:200;
Mouse anti human/mouse/rat AREG (Santa Cruz Biotechnology, #SC-74501), dilution 1:50;
Rabbit anti human/mouse/rat SEMA3A (Abcam, #AB199475), dilution 1:300;
Mouse anti human/mouse/rat SNCG (Santa Cruz Biotechnology, #SC-65979), dilution 1:50;
Alexa Fluor® 488-conjugated AffiniPure Donkey Anti-Mouse IgG (H + L) (Jackson ImmunoResearch, #715-545-150), dilution 1:250;
Cy™3-conjugated AffiniPure Donkey Anti-Rat IgG (H + L) (Jackson ImmunoResearch, #712-165-150), dilution 1:250;

Alexa Fluor® 647-conjugated AffiniPure Donkey Anti-Rabbit IgG (Jackson ImmunoResearch, #711-605-152), dilution 1:250; Biotinylated-Goat Anti-Mouse Ig (Multiple Adsorption) (BD Biosciences, #550337), dilution 1:100.

### Quantification and statistical analysis

All the bioinformatics and related statistical analyses were performed using the R packages described in corresponding method sections and figure legends. Mean fluorescence intensity of flow cytometry data was analyzed by two tailed Student's *t* test. The sample size in each experiment was described in corresponding result sections, figure legends as well as in Supplementary Data 1.

### Reporting summary

Further information on research design is available in the Nature Portfolio Reporting Summary linked to this article.

### Data availability

The raw and processed high throughput sequencing data generated in this study have been deposited and made publicly available at Gene Expression Omnibus under accession numbers GSE223153 and GSE222990. Raw read counts and metadata of referenced scRNA-seq datasets used in this study are available for download according to the authors' instructions from corresponding refs. 15,19,20,24, including GSE172380, GSE141017, [http://singlecell.charite.de/cellbrowser/pancreas/], and HTAN Data Coordinating Center Data Portal under the HTAN WUSTL Atlas [https://data.humantumoratlas.org/], respectively. GTEx normal pancreas RNA expression data can be accessed from GTEx Portal [https://gtexportal.org/home/downloads/adult-gtex#bulk_tissue_expression]. TCGA pancreatic cancer RNA expression data are available at GDC data portal [https://portal.gdc.cancer.gov/projects/TCGA-PAAD]. Human reference genome GRCh38 is available at [https://hgdownload.soe.ucsc.edu/goldenPath/hg38/bigZips/].

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

## Acknowledgements

This work is supported by the Cancer Prevention and Research Institute of Texas (R1219, P.W.), NIH/NCI (R21 CA218968, P.W., R01 CA237159, P.W.), NIDDK (R01DK110361, P.W.), and the William and Ella Owens Medical Research Foundation (P.W.). P.W. is a Cancer Prevention and Research Institute of Texas (CPRIT) scholar, and the P.W. group was supported by CPRIT. Michael Nipper was supported by a pre-doctoral fellowship through CPRIT Research Training Award RP 170345; J.L. was supported by a post-doctoral fellowship through CPRIT Research Training Award RP140105 and is currently supported by the American Cancer Society and Mays Cancer Center for the American Cancer Society Institutional Research Grant Early Access Pilot Award (IRG-21-147-01-IRG, P30CA054174). A.D. is supported by the Initiative to Maximize Student Development (IMSD) T32 training grant (T32 GM148752). The funders had no role in study design, data collection and analysis, decision to publish, or preparation of the manuscript. The authors acknowledge Dr. Seung Kim (Stanford University) for kindly providing KRAS construct. The authors thank Dr. Efsun Arda (NIH) and Dr. Zhijie Liu for their critical comments. The authors acknowledge Dr. Junichi Takagi for kindly providing engineered cell line that produced biologically active Wnt/afamin. Part of data was generated in the Flow Cytometry Shared Resource at UT Health San Antonio which is supported by the NCI grant from Mays Cancer Center (P30CA054174), the CPRIT grant (RP210126) and NIH grant (1S10OD030432-01A1). EM images were captured at UT Health San Antonio Electron Microscopy Laboratory. High throughput sequencing data described in this study was generated in the Genome Sequencing Facility/Mays Cancer Center Next-generation Sequencing Facility, which is supported by UT Health San Antonio, NIH-NCI P30 CA054174 (Cancer Center at UT Health San Antonio) and NIH Shared Instrument grant S10OD030311 (S10 grant to NovaSeq 6000 System), and CPRIT Core Facility Award (RP220662 to Y.C.). Human pancreatic tumor tissues were procured from the Mays Cancer Center Biorepository led by Dr. Robin Leach and assisted by Austen Lee. Part of the histology studies were assisted by the Histology and Immunohistochemistry Laboratory at UT Health San Antonio.

## Author contributions

Conceptualization: P.W., J.L. Data curation: J.L., Y.X., M.H.N, A.A.D., N.A., K.L., J.J.D., D.A., A.S., P.W. Formal analysis: P.W., J.L., Y.X., M.H.N., A.A.D., N.A., K.L., Z.Y., Y.C., L.-Z.S., S.Z., A.D.S., F.E.S., H.W., M.E.F-Z. Funding acquisition: P.W. Methodology: J.L., Y.X., M.H.N., N.A., K.L. Project administration: P.W. and J.L. Supervision: P.W. Writing – original draft: J.L., Y.X., M.H.N., P.W. Writing – review & editing: J.L., Y.X., M.H.N., K.L., P.W., N.A., C.C.M., Z.Y., Y.C., L.S., S.Z., A.D.S., F.E.S., H.W.

## Competing interests

The authors declared no competing interests.
