## [Peer Review File · Nature Communications]

REVIEWER COMMENTS

Reviewer #1 (Remarks to the Author):

This study reports the derivation and modification of organoid cultures from human ductal and acinar cells, followed by transcriptomic and phenotypic analyses. The system used in this study is highly novel, and the results have important implications for a lingering controversial question in pancreatic cancer biology - the cell of origin of pancreatic tumorigenesis. The experiments are well-reasoned, the interpretation is mostly sound, and the results are clearly presented. However, the manuscript as written could be improved with attention to the following points:

1. The Methods sections provides protocols for both 2D and 3D culture of pancreatic ductal cells, and the Discussion mentions as a caveat that the ductal cells were mostly cultured in 2D. However, it is not clear in the Results which experiments were performed in 2D vs 3D culture for ductal cells - this should be clarified. Also, the rationale for 2D culture of ductal cells should be better explained - methods for 3D ductal cell culture have been published by multiple groups, and the different culture approaches may confound the results of the current study.
2. More detail is needed in the Results section about the strategy for introducing mutant KRAS into the acinar and ductal organoids. Presumably this is not expressed at the endogenous level. Since increased expression of KRAS is also associated with pancreatic tumorigenesis, how might this affect interpretation of the results?
3. The comparison of TCGA PDAC data to normal pancreas in GTEX is problematic because of the differences in cell types in these tissues, i.e. GTEX normal pancreas will be mostly acinar cells while PDAC will be enriched with CAFs. I think this comparison is of questionable utility when used to interpret data from epithelium-only organoid cultures.
4. The clinical utility of the results of this study as biomarkers is likely oversold. As above, the TCGA/GTEX comparison is problematic, and it is not clear why genetically modified organoids should provide more useful clinical biomarkers than human biosamples. This part of the study should be better justified or the language toned down re: clinical utility.
5. The observation that the 3D acinar culture conditions cause phenotypic changes similar to ADM/PanIN in the absence of KRAS mutation is an important caveat to the acinar culture system. While the authors mention this briefly, I think this needs to be discussed more in depth, in particular with

references to the cell-of-origin debate. It does not negate the important findings of this study about the tumorigenic potential of acinar cells, but it provides important context to the results.

Reviewer #2 (Remarks to the Author):

In the manuscript "Reconstitution of PDAC tumorigenesis with primary human pancreatic cells 2 reveals acquisition of oncogenic transcriptomic features at tumor onset" the author established an in-vitro culture of human acinar cells and human ductal cells. They expanded sorted acinar cells to form 3D organoids and established 2D culture of ductal cells. In addition, the authors introduce oncogenic Kras and knocked out p16/CDLN2A, TP53, and SMAD4 (KPTS) in isolated acinar or ductal cells and examined the potential of these cells to form tumors in immunodeficient mice. RNA-seq was used to compare gene expression in different cell cultures and human PDAC published data. Overall the work is interesting, although tumor formation by KPTScells is not surprising, it is indeed shown for the first time. The established in-vitro system can be useful, however, according to the RNA-seq data, the cells are not similar to fresh isolated acinar or ductal cells, it is not clear if these cells are similar to metaplastic cells or cancer cells, and whether they are heterogeneous. Major related points are listed below:

- 1) According to the gene expression profile shown in Figure 1, the cultured acinar cells acquire mixed acinar and ductal programs and therefore do not faithfully mimic either of the primary cell types. In Figure 1f they compare specific genes to previous works that profiled metaplastic cells in mice. Full gene expression comparison including size effects is missing. To what extent cultured acinar cell gene expression overlap with each metaplastic cell type?
- 2) Related to point 1 the author should perform scRNA-seq of cultured acinar cells, it is critical to test if the cells are heterogeneous, and if this is the case sorted subpopulations can help in investigating each metaplastic cell type, alternatively, the reader should be noted if each cell expresses a mix of the metaplastic cells programs.
- 3) The acinar-KPTS cells and the ductal-KPTS cells expressed genes that mark "Duct-lik1", "Duct-lik2", PanINs, and PDAC, but here as well, it is critical to know if all the culture cells express all the programs of whether the cultured include cells at several different states. The authors' conclusion that Acinar-KPTS cells are in transition from pain to PDAC is highly speculative and scRNA-seq of the cultured acinar and ductal cells is needed.
- 4) In the data that is shown in Figure 3 the author includes an analysis that is based on cells from several cultures that include more than one donor. What is the heterogeneity between cells that originate from different donors and between different cultures from the same donor?

5) The reduction in MHC-I and especially in MHC-II related expressed genes in the KPTS model is interesting. The author should repeat these experiments using relevant antibodies for MHC molecules. This is needed to confirm the relevance of this result to tumor biology.

6) The list of genes in Figure 6f may assist in early detection but the reader is left with a list of genes without any follow-up experiments. The clinical relevance or mechanistic investigation of at least some of these genes is needed.

7) The new culturing method has the potential to be beneficial for the community but it is challenging to achieve fresh human pancreatic tissue. The author should show if the protocol can be used to culture mouse primary.

Reviewer #1

Comment 1. The Methods sections provides protocols for both 2D and 3D cultures of pancreatic ductal cells, and the Discussion mentions as a caveat that the ductal cells were mostly cultured in 2D. However, it is not clear in the Results which experiments were performed in 2D vs 3D culture for ductal cells - this should be clarified. Also, the rationale for 2D culture of ductal cells should be better explained - methods for 3D ductal cell culture have been published by multiple groups, and the different culture approaches may confound the results of the current study.

Response: *Thanks for this comment. We initially attempted to culture both acinar and ductal cells as 3D organoids using same protocol as reported (reference 5) (Fig. 1C). However, the ductal cells could not survive in 3D culture for more than 3-4 passages, which made it difficult for our investigations. We managed to establish several 2D cultured ductal cell lines whose lineage identity was confirmed by methylation profiling (Fig. 2). Thus, all the experimental results involving cultured/modified ductal cells (described in Figs 2-6), were generated from these 2D cultures. In the manuscript, any ductal samples not labeled as “fresh” were all derived from 2D culture. The manuscript has been carefully revised to make this clear in both the Results and Methods sections.*

We are aware of the reports regarding 3D ductal cultures (e.g., references 5 and 38). These reports determined ductal identity by examining the expression of certain acinar and ductal markers (e.g. AMY, PTF1a, Ck19, SOX9, etc) in organoids. However, the expression of acinar markers could be rapidly downregulated while ductal markers could be induced in acinar cells under stress conditions. This was also observed in our acinar organoids, as shown in Fig. 1D and Extended Fig. 1G. Therefore, determining ductal identity of cultured organoids in these reports solely based on the expression of certain markers is questionable. In comparison, here we firstly flow-sorted two lineages, followed by comprehensive DNA methylation profiling (Fig. 2 and Extended Fig. 2) integrated with RNA-seq data, to confirm the lineage identity in our culture system. We believe the observations from our model system better represent lineage phenotype. We have revised the Discussion section to address this.

Comment 2. More detail is needed in the Results section about the strategy for introducing mutant KRAS into the acinar and ductal organoids. Presumably this is not expressed at the endogenous level. Since increased expression of KRAS is also associated with pancreatic tumorigenesis, how might this affect results' interpretation?

Response: *As suggested, we have edited the Result section to briefly describe the delivery/expression of oncogenic KRAS into cultured cells. Details on lentivirus vector construction and infection procedure are also described in the Method section.*

Surprisingly, the overexpression of oncogenic KRAS induced only minimal transcriptional and phenotypic changes (Fig. 3A-B). In addition, we were unable to generate xenograft tumors when transplanting cells that only overexpressed

oncogenic KRAS, without additional mutations (e.g., p16, p53 and SMAD4). Therefore, in our model system, the overexpression of oncogenic KRAS appears to have a limited effect and it is insufficient to induce transformation. This information was mentioned in the original version of the Result section. We have also revised the Discussion section to elaborate on the limited impact of oncogenic KRAS in our model.

Comment 3. The comparison of TCGA PDAC data to normal pancreas in GTEX is problematic because of the differences in cell types in these tissues, i.e. GTEX normal pancreas will be mostly acinar cells while PDAC will be enriched with CAFs. I think this comparison is of questionable utility when used to interpret data from epithelium-only organoid cultures.

Response: *Thanks for this thoughtful comment. Indeed, TCGA PDAC samples have substantial amount of stromal and immune cell composition, as previously reported (reference 26). This is precisely one of our rationales to use the findings from our model system to refine the comparison between TCGA PDAC and GTEX normal pancreas data. Such integration of our epithelial only data with TCGA/GTEX comparison will help to filter out interference from stromal/immune contamination, while keeping the observation clinically relevant. We now added this information in the Results section.*

In addition, for the data analysis, we also included only the “high purity” TCGA samples as identified in abovementioned (reference 26), the results turned out to be very similar. Thus, to keep the manuscript concise and adhere to the word limit, we decided to include all TCGA PDAC samples in our data analysis.

Comment 4. The clinical utility of the results of this study as biomarkers is likely oversold. As above, the TCGA/GTEX comparison is problematic, and it is not clear why genetically modified organoids should provide more useful clinical biomarkers than human biosamples. This part of the study should be better justified or the language toned down re: clinical utility.

Response: *Thanks for the suggestion. As the reviewer pointed out, the comparison between TCGA PDAC and GTEX normal pancreas suffers from the presence of stromal/immune contamination. Therefore, we employed our model system to refine the potential biomarker list, aiming to minimize such interference. Secondly, most human PDAC samples are at advanced stages, while our model represents early tumor initiation stage. Revealing transcriptomic changes in early PDAC is a crucial step towards discovering potential markers for early diagnosis and therapeutic targets. These are some of the unique advantages of our model system presented in this work. In addition, we have now performed follow-up IHC staining in our tumor samples as well as human PDAC samples, validating a subset of candidate biomarkers found in our model system in these samples. Result showed that the tested genes are consistently highly expressed in tumors, with no or limited expression in adjacent normal tissues, confirming their potential clinical relevance. Representative IHC*

images are now incorporated as **new Fig. 6F-I** and **Extended Fig. 6A**. The Results and Methods section has been revised accordingly.

Comment 5. The observation that the 3D acinar culture conditions cause phenotypic changes similar to ADM/PanIN in the absence of KRAS mutation is an important caveat to the acinar culture system. While the authors mention this briefly, I think this needs to be discussed more in depth, in particular with references to the cell-of-origin debate. It does not negate the important findings of this study about the tumorigenic potential of acinar cells, but it provides important context to the results.

Response: *Thanks for the comment. As shown in **Fig. 1E** and **Fig. 3B**, the wild type acinar cultures, in the absence of KRAS mutation, already became metaplastic and acquired a disease-associated signature, revealing the potential acinar origin of these disease-associated cell populations emerged during inflammatory process (e.g., pancreatitis) and PDAC development. This observation may be attributed to 1) the acinar cells are highly plastic and prone to phenotypic change under in vitro culture stress, and 2) the supplementation of EGF in the growth media could partially activate the oncogenic KRAS pathway.*

In addition, as mentioned in Response 2, expression of oncogenic KRAS into the acinar culture did not cause significant transcriptomic changes in these cells which had already become metaplastic. We postulate that oncogenic KRAS in acinar cells may be an essential prerequisite, rather than a sufficient driver, for PDAC transformation in our model system. This aligns with a previous report (reference 16), demonstrating that oncogenic KRAS in mouse acinar cells can lock an inflammation-induced oncogenic network to allow acinar transformation, rather than directly initiating the transformation. We have now revised the Discussion section accordingly.

Reviewer #2

Comment 1. According to the gene expression profile shown in Figure 1, the cultured acinar cells acquire mixed acinar and ductal programs and therefore do not faithfully mimic either of the primary cell types. In Figure 1F they compare specific genes to previous works that profiled metaplastic cells in mice. Full gene expression comparison including size effects is missing. To what extent cultured acinar cell gene expression overlap with each metaplastic cell type?

Response: *Thanks for the comment. While our isolated fresh acinar and ductal cells showed distinct lineage features (Fig. 1A, B, D, and Extended Fig. 1A-C), indeed, after long term culture, the acinar organoids lost lineage identity and partially acquired ductal-like characteristics. This observation is consistent with the previous notion that acinar cells are highly plastic under stress conditions, and can readily undergo acinar-to-ductal metaplasia. Therefore, we believe that our acinar organoids faithfully mimic the phenomena occurring in acinar cells under challenging environment.*

As suggested, we have now performed full gene expression PCA analysis, incorporating our RNA-seq samples and all reference datasets described in Fig. 1E. The result showed a great similarity between our cultured acinar organoids and certain metaplastic populations identified in the corresponding references, namely chief-like cells, pit-like cells and mucin5B/ductal cells. The result of PCA analysis is now incorporated as new Fig. 1F. We have also revised the Method and Result sections to describe the new analysis and findings.

Comment 2. Related to point 1 the author should perform scRNA-seq of cultured acinar cells, it is critical to test if the cells are heterogeneous, and if this is the case sorted subpopulations can help in investigating each metaplastic cell type, alternatively, the reader should be noted if each cell expresses a mix of the metaplastic cells programs.

Response: *This is a great point. Investigating into the heterogeneity of acinar cultures carries disease significance. However, undertaking scRNA-seq as well as subsequent data analysis requires a considerably extended timeline and additional resources beyond our current capacity. Also, we believe that these studies extend beyond the primary focus of our current manuscript.*

As such, with limited resources, instead of performing scRNA-seq experiment, we attempted to address this comment by performing a bulk RNA deconvolution analysis using Multi-Subject Single Cell deconvolution method (MuSiC, reference 21), a well-established algorithm designed to infer the cell type composition present in bulk RNA-seq samples by using a characterized scRNA-seq dataset as reference (reference 19). Our deconvolution analysis estimated approximately 90% of the isolated acinar cells to have normal acinar cells phenotype, thereby showcasing the robustness of such analysis. Using this analysis, we found that around 90% of our cultured acinar cells align with the metaplastic chief-like acinar cells, rather than any other cell types documented in the reference dataset.

*While we recognize that computational analysis cannot replace experimental evidence, this analysis indicates that our cultured acinar cells likely resemble a specific metaplastic cell population with limited heterogeneity. The results of the deconvolution analysis have been incorporated as **new Extended Fig. 1H**, we also updated both the Method and Results sections accordingly to describe the methodology and findings of our analysis.*

Comment 3. The acinar-KPTS cells and ductal-KPTS cells expressed genes that mark "Duct-lik1", "Duct-lik2", PanINs, and PDAC, but here as well, it is critical to know if all the culture cells express all the programs of whether the cultured include cells at several different states. The authors' conclusion that Acinar-KPTS cells are in transition from pain to PDAC is highly speculative and scRNA-seq of the cultured acinar and ductal cells is needed.

Response: *Thanks for this great suggestion. As noted in response 2, we would be excited to delve into the heterogeneity of metaplastic and oncogenic cell populations given sufficient resources. With limited capacities at the moment, we performed deconvolution analysis using MuSiC package in R software to infer the cell type composition in bulk RNA-seq samples using treatment naive human PDAC scRNA-seq samples as a reference (dataset from reference 24). As shown in the **new Fig. 4B**, our analysis inferred that over 99% of our fresh acinar cells have a normal acinar phenotype and around 80% of our fresh ductal cells as duct-like 1 cells present in the reference dataset. This data again underlines the robustness of such analysis.*

Further, from this analysis, we found that around 85% of our cultured metaplastic acinar organoids were inferred as duct-like 2 cells, categorized as a disease associated metaplastic population in the cited study. Interesting, nearly 100% of our acinar-KPTS cells were inferred as PDAC cell population present in the reference scRNA-seq samples, while our ductal-KPTS samples were inferred to contain a small fraction of PanIN and duct-like 1 cells.

*The transcriptional similarity of our KPTS cells to the treatment naive human PDAC cells confirms the clinical relevance of our model system. In addition, given the prevalent PanIN histology prominently observed in our acinar-derived tumors, it appears highly plausible that our acinar-KPTS cells represent a transitional stage progressing from PanIN to a fully developed PDAC. We have now added the deconvolution result as **new Fig. 4B**, and revised the Result section accordingly.*

Comment 4. In the data that is shown in Figure 3 the author includes an analysis that is based on cells from several cultures that include more than one donor. What is the heterogeneity between cells that originate from different donors and between different cultures from the same donor?

Response: *Thanks for the thoughtful comment. As recommended, we determined the heterogeneity of our samples from different donors versus different cultures by performing PCA analysis encompassing fresh, cultured, and genetically modified cultures featured in this work. The result showed that all the samples clustered*

*according to their respective states of culture, as opposed to different donor origins. This result has been now added as **new Fig. 3E**. We also revised the Method and Result section to reflect these additions.*

Comment 5. The reduction in MHC-I and especially in MHC-II related expressed genes in the KPTS model is interesting. The author should repeat these experiments using relevant antibodies for MHC molecules. This is needed to confirm the relevance of this result to tumor biology.

Response: *As suggested, we performed additional experiments in paired wild type acinar and acinar-KPTS cells by flow cytometry analysis using MHC-I and MHC-II antibodies and corresponding isotype controls. The results showed a significant down-regulation of MHC-I expression in acinar-KPTS cells compared to the paired wild type acinar samples. However, we could not detect MHC-II protein expression either by flow cytometry or Western blot. We have now added the results of MHC-I expression as the **new Fig. 5D**. We also revised the Methods and Results sections accordingly.*

Comment 6. The list of genes in Figure 6F may assist in early detection but the reader is left with a list of genes without any follow-up experiments. The clinical relevance or mechanistic investigation of at least some of these genes is needed.

Response: *We agree with this reviewer's comment. To define the disease relevance of candidate biomarkers, we first confirmed the expression pattern of selected genes in KPTS tumor and human PDAC tissues and adjacent normal tissues by IHC. Result showed that the tested genes are consistently highly expressed in tumor area, with no or limited expression in adjacent normal tissues, confirming their clinical relevance. Noteworthy, AHNAK2, a protein which was previously proposed as a PDAC prognosis marker, was positive in tumor area of 25 out of 28 tested PDAC patients, compared with only 1/28 patients positive at adjacent normal area. AREG, a member in EGF signaling which negatively correlates with PDAC prognosis, was positively expressed in tumor area of 18/28 patients, compared with 9/28 patients positive at adjacent normal area. These data confirmed the high expression profile of tested genes in both our early PDAC-like cells as well as PDAC clinical samples, demonstrating their potential as biomarkers for PDAC early detection. Representative IHC images are now incorporated as **new Fig. 6F-I** and **Extended Fig. 6A**. The Method and Result sections are also revised accordingly.*

Comment 7. The new culturing method has the potential to be beneficial for the community but it is challenging to achieve fresh human pancreatic tissue. The author should show if the protocol can be used to culture mouse primary.

Response: *Thanks for the suggestion. Given that lineage tracing is not possible in human, it has been challenging to investigate lineage-specific features which has led to the controversy on the identity of the cell of origin of human PDAC. To address this issue, here we established a model for human primary acinar and ductal cultures that allows lineage-specific studies of human PDAC. Thus, we did not find it necessary to*

use our protocol for mouse primary cultures to support the main conclusions of the study. We hope this response suffice to address this reviewer's comment.

REVIEWERS' COMMENTS

Reviewer #1 (Remarks to the Author):

The revised manuscript effectively addresses the concerns raised in the initial submission.

Reviewer #2 (Remarks to the Author):

The authors made a significant effort, improved the manuscript, and addressed all the issues that I raised.

I think it is still a question of to what extent the in vitro model recapitulates the malignant process in patients. In addition, the fact that the starting materials should be provided from the pancreas of healthy individuals may pose a significant challenge for using the new methods that are described in the paper for other scientists.

Reviewer #1:

Comment. The revised manuscript effectively addresses the concerns raised in the initial submission.

Response: *We appreciate the effort from this reviewer for reviewing our manuscript.*

Reviewer #2:

Comment. The authors made a significant effort, improved the manuscript, and addressed all the issues that I raised. I think it is still a question of to what extent the in vitro model recapitulates the malignant process in patients. In addition, the fact that the starting materials should be provided from the pancreas of healthy individuals may pose a significant challenge for using the new methods that are described in the paper for other scientists.

Response: *Thanks for this comment. We appreciate the reviewer's concern regarding how well the in vitro disease model can recapitulate the malignant process in patients. Therefore, our ongoing work is to perform single-cell RNA seq on cultured organoids and xenograft tumors to experimentally compare our model with clinical samples to further validate the clinical relevance. In addition, in our future work, we plan to perform orthotopic transplantation using humanized mice to investigate the PDAC development in a tumor microenvironment more relevant to clinical samples. Considering the current lack of an early human PDAC model, we reasoned that engineering normal human primary pancreatic cells will offer a unique opportunity to decipher the human PDAC initiation and early progression, which is the first step towards the goal of developing prevention and early diagnosis strategies for improving outcomes of this disease. We revised the discussion section to describe this limitation.*

We also acknowledge that normal human pancreas tissue samples may not be readily available. However, we described a commercial source for obtaining such samples in our Methods section. We hope this can address the reviewer's concern.